# Eruption of ultralow-viscosity basanite magma at Cumbre Vieja, La Palma, Canary Islands

Jonathan M. Castro [1✉] & Yves Feisel [1✉]

The viscosity of magma exerts control on all aspects of its migration through the crust to eruption. This was particularly true for the 2021 eruption of Cumbre Vieja (La Palma), which produced exceptionally fast and fluid lava at high discharge rates. We have performed concentric cylinder experiments to determine the effective viscosities of the Cumbre Vieja magma, while accounting for its chemistry, crystallinity, and temperature. Here we show that this event produced a nepheline-normative basanite with the lowest viscosity of historical basaltic eruptions, exhibiting values of less than 10 to about 160 Pa s within eruption temperatures of ~1200 to ~1150 °C. The magma's low viscosity was responsible for many eruptive phenomena that lead to particularly impactful events, including high-Reynolds number turbulent flow and supercritical states. Increases in viscosity due to crystallization-induced melt differentiation were subdued in this eruption, due in part to subtle degrees of silica enrichment in alkaline magma.

[1] Institute of Geosciences, University of Mainz, Becherweg 21, Mainz D-55099, Germany. ✉email: castroj@uni-mainz.de; yfeise02@uni-mainz.de

Of magma's many physical properties, its viscosity exerts the most profound influence on the way it moves and erupts. From ascent, to bubble growth, to fragmentation[1,2] and the emplacement of lava flows[3], a magma's physical evolution with its surroundings depends on the way it deforms and on the rate of that deformation. Hence, measuring viscosity and relating it to antecedent eruptive processes can greatly improve our understanding and eventual modeling of volcanic activity. Viscosity is however a complicated function of many variables, including, temperature, crystal and bubble content, and melt composition, each of which changes with time during the course of the eruption[4–7]. Despite these complexities, and in contrast to other intensive properties like temperature, viscosity can be summarily assessed during eruptions with the naked eye, by for example observing how fluid the lava appears and estimating flow velocities as it surfaces[8–10]. Indeed, some of the Earth's most fluid lava flows were also observed to be the fastest, and many of these were fueled by eruptions of alkaline mafic magma[9,11,12]. Magmas of alkaline affinity, including basanites, ankaramites, and nephelinites[13], have characteristically low-SiO$_2$ content (<45 wt.%), which fosters particularly high fluidity and concomitant velocities ($\sim$10–20 m s$^{-1}$) that are seldom seen during eruptions of tholeiitic magma[14,15]. Perhaps most importantly—and from a hazards perspective—alkaline magma's low apparent viscosities[12,16] make hazard mitigation and evacuation difficult as these lavas may rapidly inundate vast tracts of land, destroying infrastructure, and in extreme cases, causing the tragic loss of life[17]. Adequate crisis response to eruptions of fluid alkaline magma must account for their potentially enhanced threat brought on by their low viscosities.

Our aim is to quantify the eruptive viscosity of the most recent and highly impactful eruption of alkaline magma, that of the 2021 basanite eruption of Cumbre Vieja, La Palma[18]. The eruption captured the attention of the world because of the profound human and infrastructural impact it has had on the Canarian island and due to the spectacular styles and rates at which volcanic activity occurred (Fig. 1). Much of this activity was documented by the Canary Government Public Television[19] (RTVC), which provided nearly continuous video feeds from different cameras aimed at the vent and emerging lava streams. Such videos show rapid lava exit velocities exceeding 10 m s$^{-1}$ (Fig. 2), hydraulic jumps in cascading lava streams (Fig. 3), and the remarkable development of an expansive lava flow field that took just weeks to form (Fig. 1). All of these phenomena are the hallmarks of highly fluid lava rheology. The exceptional documentation of this eruption presents a unique opportunity to link measurable physical properties of magma—e.g., viscosity—to its natural flow behavior. We have determined the anhydrous bulk viscosity of La Palma's Cumbre Vieja magma using natural ash samples and a Couette flow device across a range of volcanologically relevant eruptive temperatures (1110–1300 °C) in order to establish a rheological context for these impacts. As will be seen in the sections below, our results show that the viscosity of Cumbre Vieja lava could have been exceedingly low (tens of Pa sec) across a range of permissible eruption temperatures. These physical properties position the Cumbre Vieja basanite in a rarely observed behavioral class, in which inertial effects on the flow were important, suggesting that some lava flow facies were probably turbulent and subject to hydraulic jumps[20,21].

## Results

### Geology and Progression of the 2021 Eruption of Cumbre Vieja, La Palma.
The island of La Palma is part of the Canary Island archipelago, located off the west coast of Africa (Fig. 1). The island chain comprises seven emergent seamount volcanoes whose origin is thought to be related to mantle plume activity[22]. Recent ($\sim$125 ka to present) magmatism on La Palma[23,24] has occurred along the Cumbre Vieja ridge, an elongate rift-controlled massif comprising abundant volcanic and plutonic rocks of alkaline magma affinity[25]. Particularly abundant amongst Cumbre Vieja's most recent eruptions are lavas and tephras of basanite-tephrite composition[26]; these are characterized by low SiO$_2$ contents (<45 wt.%) and low to moderate total alkali concentrations[27] ($\sim$4–8 wt.%). Basanite lava flows emplaced during these historical eruptions (e.g., the 1677 San Antonio and 1971 Teneguía events) are notable for their observed fluid behavior[24] and resemble those of the 2021 event in that they are long (3–8 km) and slender, characteristics of which manifest both magma transport properties (e.g., viscosity) and direct emplacement paths to the sea (Fig. 1).

The 2021 eruption of Cumbre Vieja began on 19 September at 14:10 UTC from several closely spaced vents defining a NW-SE oriented fissure, located approximately 4 kilometers SSE of El Paso[28]. The event was preceded by about one week of elevated seismic activity and ground deformation. The eruption progressed from an early explosive phase that sent ash clouds to over 5000 m elevation, and within hours the vents started effusing highly fluid, rapidly advancing lava flows alongside continued fire fountaining and violent strombolian activity. Within 24 hours of the eruption's onset, a 3 km lava flow had already been emplaced[28] (Fig. 1), and a few days later these lavas began inundating the town of Todoque. It took just under ten days for the lavas to make their first of three ocean entries, some 5.5 km due west of the volcano.

In total, the 2021 eruption lasted over 85 days, exceeding the 1585 eruption of Tehuya[29] (a.k.a. Jedey) by about one day. Owing to continued pyroclastic venting during the protracted effusive activity, a massive pyroclastic cone was built, itself approximately 225 m in height and 1125 m above sea level[28] (Fig. 1b). Throughout the whole eruptive duration, the volcanic cone hosted particularly complex and frequently shifting eruptive dynamics over short time scales. Seemingly hourly-to-daily fluctuations in eruption style (explosive vs. effusive), lava and pyroclast fluxes—manifested by variable fire fountain height—in addition to intense local and diffuse degassing patterns, were notable across the many closely spaced active vents within and on the flanks of the pyroclastic cone. Moreover, the constellation of active vents expanded on many occasions (e.g., 2 October, 18 November, 25 November)[30] to include new points of lava effusion and fountaining. As the formation of these points-of-emission are spatially linked to sub-surface supply of magma, we will focus the descriptions of eruptive phenomena that occurred close to the active vent during the week of our sampling campaign (14–20 November 2021), and which provide a rare look at how exceptionally fluid the newly emergent lava was.

Our observations of Cumbre Vieja's near-vent activity on 18 November 2021 indicate that at 20:23 local time, two new eruptive vents formed on a bench positioned several tens of meters below the cone's summit (Fig. 2). Due to night-time viewing conditions, the lava emerging from these vents was extremely bright, and emitted a white radiance in contrast to the duller, orange-incandescent color of lava effusing from the south side of the cone and the ongoing lava fountains erupting directly above the new vents at that time. The bubbling vents, which themselves exhibited mild fountaining activity to heights of about 10–30 m, appeared to overflow down a steep west-facing rampart, which in turn created two streams of rapidly moving, highly fluid lava (Fig. 2; Supplementary Information). From our vantage point in El Paso, we could see that the lava in these cascades flowed for a few tens of meters down a relatively gentle slope that later gave way to a steeper descent with a length of

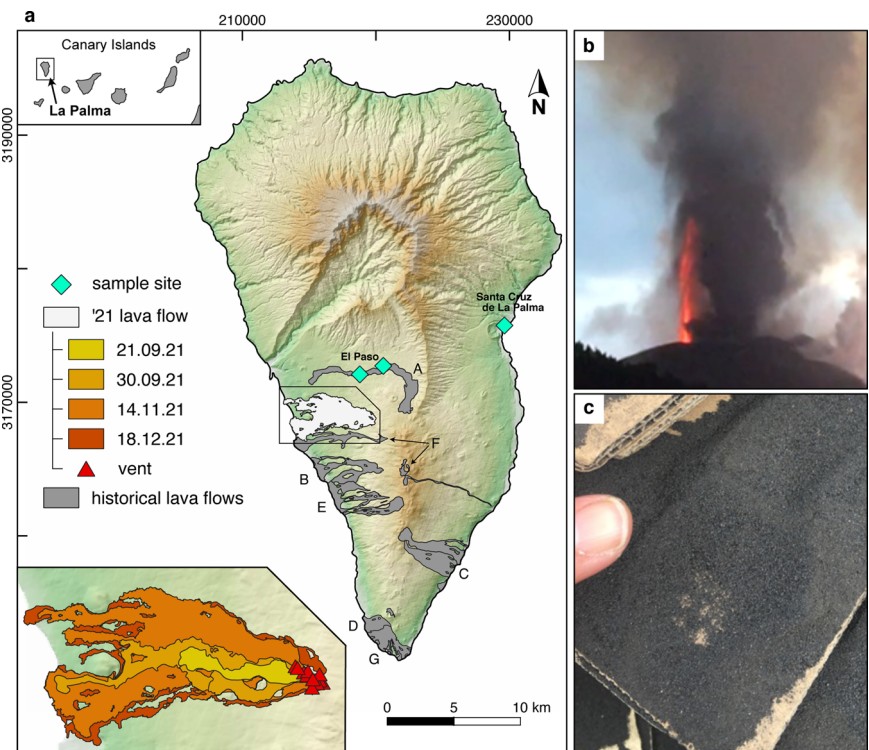

**Fig. 1 Overview of the geological setting, eruptive behavior, and ash sampling at Cumbre Vieja. a** Map of La Palma, Canary Islands showing the locations of the September 2021 eruptive vents, (El Paso), along with the resultant lava flows. Historic lava and tephra eruptions of La Palma are also shown as gray colored zones and correspond to the Mtña. Quemada (A; 1430/1440), Jedey (B; 1585 CE), Martin (C; 1646 CE), San Antonio (D; 1677), El Charco (E; 1712), San Juan (F; 1949), and Teneguia (G; 1971) eruptions. **b** This study utilized fresh basanite ash produced during the 14–20 November explosive activity, which comprised fire fountains (500–2000 m) that brought coarse ash to the points of sampling, shown with diamond symbols. Ash was collected within hours of being deposited on flat surfaces (**c**). The map (frame **a**) uses data of the Copernicus GLO-30 Digital Elevation Model (European Space Agency, Sinergise 2021). Lava flow data of the 2021 eruption of Cumbre Vieja was provided by the Copernicus Emergency Management Service (European Union 2021), EMSR546. The data of historical lava flows is based on the MAGNA 50 geological map provided by the Instituto Geologico y Minero de Espana via Web Map Service and is modified after refs. [18] and [25].

about fifty meters. Thus, the twin lava streams appeared to gain momentum as they progressed downslope. RTVC video at the same time captured the lava streams' motion and structural development in greater detail (Fig. 2), and show that as lava traveled down the slope, several bright arcuate structures would periodically form and persist the entire length of the slope, even as the underlying ramp changed in its lateral (width) dimension. The bluntness and brightness of these lava pulses increased as the flow gained momentum, approximating a series of surging waves downslope. Towards the end of the visible part of the cascades (Fig. 2), the bright arcuate structures coalesced, like multiple water drops flowing down a sloping pane of window glass. In this way, the pulses of descending lava behaved like low-viscosity drops running down previously wetted trails from the prior passage of fluid.

In addition to our field observations indicating peculiar lava effusion behavior of highly fluid near-vent magma (Fig. 2), several instances of undulose wave-like structures were present in RTVC videos of near-vent lava emerging from vents on the cone's southern flank (Fig. 3 depicting activity on 25 November 2021). By our estimation, and based on the fact that these were stationary, long-lived (minutes) waves that did not move significantly down stream, these reflect the formation of standing waves in the flows. Standing waves have been documented in other recent basaltic eruptions, including the 2018 eruption of Kīlauea, USA[14,21]. A lack of scale information in the videos prevents us from describing these surface waves further, however, their presence alone is a strong indication of supercritical flow in lavas[20,21].

**Fresh ash collection from the erupting Cumbre Vieja massif.** Due to an exclusion zone set up around the volcano by civil protection authorities, we were not able to obtain lava samples for use in rheology experiments. However, abundant tephra deposits were sampled near the cities of El Paso and Santa Cruz de La Palma, during the week of 14–20 November 2021 (Fig. 1). The activity at this time was visually monitored from the points of sampling, and much like the activity that dominated the whole 3-month episode, consisted of: (1) persistent ash venting and lava fountaining, (2) effusive activity sourced by 5–8 different vents, (3) profuse emission of gas from both eruptive vents and from the flanks of the cone (Fig. 1). All samples used in experimental and analytical work were obtained shortly after they were erupted, from locations to the northwest of the active cone, just outside of the exclusion zone and roughly 3–4 km distant from the vent, and from distal areas near the capitol city of Santa Cruz de La Palma (Fig. 1). The samples were deposited as coarse ash to fine lapilli (~0.5–5 mm) grains on flat surfaces (e.g., cardboard sheets) that were arranged in the field for collection. Tephra was collected in 1 h intervals—the amount of time needed for the particles to obscure the flat surface—as the material rained out from the eruption plume. We used a brush to push the pyroclasts into air and water-tight plastic bags. In total, approximately 5 kg of tephra was collected during the week of observation. The newly lain ash was examined under a binocular microscope to ensure that no paper or other contaminants entered the material. In addition, several coarse ash (~2 mm) to fine lapilli grains (3–5 mm) were collected from areas near the

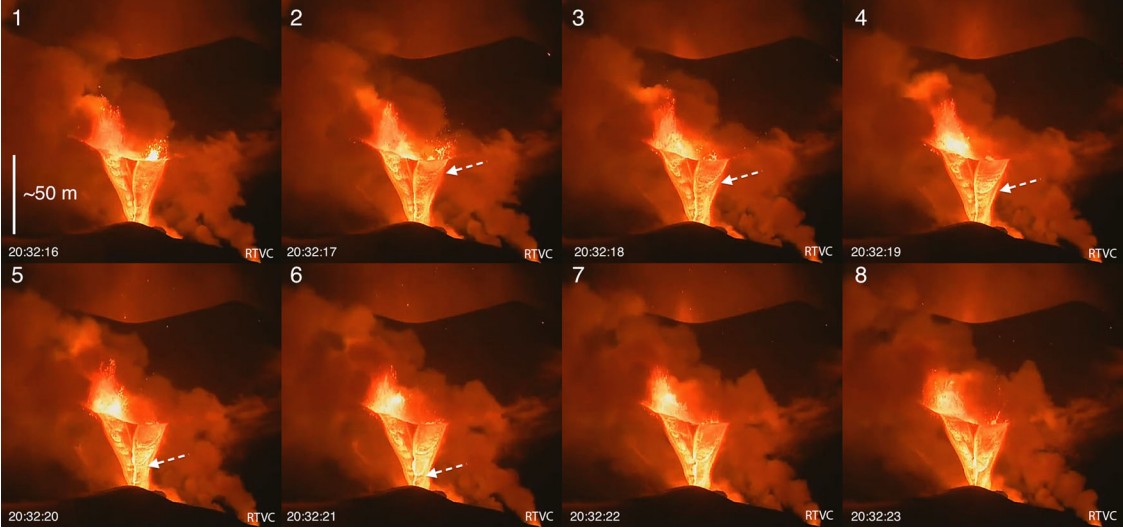

**Fig. 2 Cascading lava streams at Cumbre Vieja, 18 November, 2021.** Montage of image frames (1–8) collected from an open-source video feed (RTVC/TV-Canarias) of the eruption on 18 November 2021, at approximately 20:32 h local time. These images depict twin "lava cascades" that erupted over a period of about 30 min, and were sourced from two newly formed (~20:23 local time) and closely spaced, fountaining vents positioned atop a steep crater wall. The eruptive cone's peak is visible in the background. The series of images progress chronologically from left to right in each row, providing a short (~7 s) timeframe of this activity, so as to highlight features of the highly fluid, freshly emergent lava. The white arrows indicate the formation and displacement of one of several blunt arcuate flow fronts that formed mid-channel and whose progression downslope caused an intensifying of the incandescent fringe that demarcates the flows' fronts and margins. These features—the blunt fronts—formed in pulses every 1–3 s and maintained their form despite flowing on previously wetted lava channels. The last two frames (7 and 8) depict the formation and progression of a new front. This behavior resembles that of water-flow phenomena including bores and swash waves in that a distinct discontinuity or hydraulic jump is maintained[51–53]. Such repetitive flow lobe activity has not been observed in other parts of the flow field and is likely a consequence of the ultralow viscosity and fast downward flow of proximal lava. These video data were furthermore used to determine advance rates (to ~12 m s$^{-1}$).

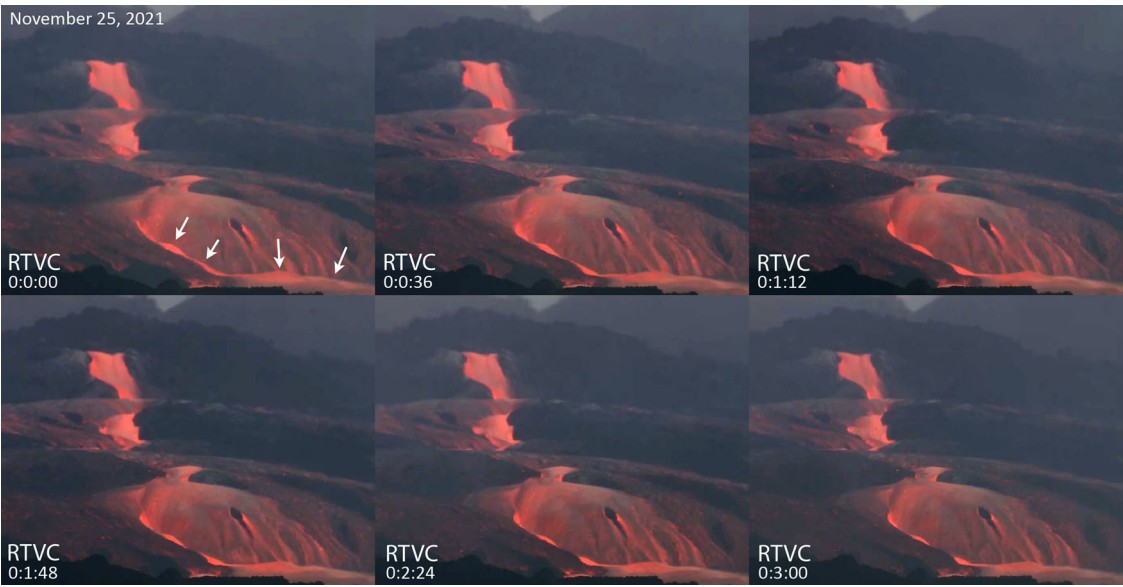

**Fig. 3 Supercritical flow phenomena in near-vent lavas, Cumbre Vieja.** Montage of image frames collected from open-source eruption video feed (RTVC/TV-Canarias) indicating the existence of standing waves (white arrows) in lavas of the near-vent region on 25 November 2021, at approximately 9:26 a.m. local time. The exact location and scale of these waves are unknown. The time markers show relative time for a three-minute interval of the sampled video. The presence of standing waves qualitatively indicates that this flow was in a supercritical state, involving high Froude numbers[20]. White arrows indicate the positions of a wave train that remained in place for a total of 8 min. After this period, the waning of the mass discharge rate at the vent, and crust formation ultimately lead to the disappearance of these standing waves.

sampling sites, embedded in epoxy, and polished for electron probe microanalysis and imaging by back-scattered electron methods. These grain separates derive from the same bulk samples used for both XRF analyses and rheometry experiments.

**Geochemistry and mineralogical characteristics of eruption products.** The major and trace element composition of tephra samples was determined by XRF (Table 1). These data indicate that the magma is a basanite (Fig. 4), corroborating the results of

**Table 1 Major element compositions and thermo-rheological properties of basanite from the 2021 eruption of Cumbre Vieja, La Palma.**

**GEOCHEMISTRY**

| | Tephra XRF | Tephra 1-3 | Tephra 4 | Glass Inclusions | CPX Tephra 1-3 | | CPX Tephra 4 |
|---|---|---|---|---|---|---|---|
| Component (wt.%)[a] | bulk ($n = 2$) | matrix gl. ($n = 64$) | matrix gl. ($n = 17$) | ($n = 6$) | rim + matrix ($n = 8$) | core ($n = 9$) | rim + matrix ($n = 7$) |
| $SiO_2$ | 43.7 (0.3) | 46.0 (0.2) | 44.1 (0.3) | 43.35 (0.84) | 45.7 (1.3) | 46.6 (2.0) | 45.7 (1.5) |
| $Al_2O_3$ | 14.1 (0.3) | 16.5 (0.2) | 15.1 (0.1) | 16.14 (0.86) | 7.5 (1.0) | 6.8 (1.7) | 6.8 (1.4) |
| $Fe_2O_3$ | 13.4 (0.2) | n.d. | n.d. | n.d. | n.d. | n.d. | n.d. |
| FeO | n.d. | 11.4 (0.2) | 12.6 (0.2) | 11.36 (0.3) | 7.6 (0.4) | 7.3 (0.4) | 8.2 (0.5) |
| MnO | 0.19 (0.01) | 0.21 (0.03) | 0.20 (0.03) | 0.19 (0.03) | 0.15 (0.04) | 0.15 (0.02) | 0.15 (0.02) |
| MgO | 7.35 (0.04) | 3.98 (0.07) | 4.50 (0.03) | 4.92 (0.37) | 12.8 (0.5) | 13.1 (1.1) | 12.6 (0.9) |
| CaO | 11.1 (0.6) | 9.4 (0.1) | 11.0 (0.2) | 11.89 (0.91) | 22.4 (0.4) | 22.3 (0.2) | 21.9 (0.6) |
| $Na_2O$ | 3.7 (0.3) | 5.8 (0.1) | 4.8 (0.1) | 4.68 (0.46) | 0.54 (0.06) | 0.6 (0.1) | 0.5 (0.2) |
| $K_2O$ | 1.6 (0.2) | 2.46 (0.05) | 1.74 (0.07) | 1.81 (0.15) | 0.01 (0.01) | 0 | 0.04 (0.06) |
| $TiO_2$ | 3.69 (0.05) | 3.77 (0.08) | 4.27 (0.06) | 3.89 (0.28) | 3.3 (0.6) | 2.9 (0.8) | 3.3 (0.7) |
| $P_2O_5$ | 0.77 (0.02) | 1.31 (0.04) | 1.06 (0.13) | 1.1 (0.25) | n.d. | n.d. | n.d. |
| $Cr_2O_3$ | 0.09 (0.07) | n.d. | n.d. | n.d. | 0.2 (0.1) | 0.14 (0.07) | 0.12 (0.11) |
| NiO | 0.03 (0.02) | n.d. | n.d. | n.d. | 0.02 (0.03) | 0.01 (0.01) | 0.02 (0.03) |
| $SO_3$ | 0.04 (0.04) | 0.10 (0.03) | 0.07 (0.05) | 0.73 (0.4) | n.d. | n.d. | n.d. |
| LOI | 0.71 (0.09) | n.d. | n.d. | n.d. | n.d. | n.d. | n.d. |
| Cl | n.d. | 0.08 (0.01) | 0.06 (0.01) | 0.07 (0.01) | n.d. | n.d. | n.d. |
| F | n.d. | 0.12 (0.02) | 0.08 (0.01) | 0.09 (0.02) | n.d. | n.d. | n.d. |
| Total | 98.99 (0) | 101.0 (0.5) | 99.6 (0.3) | 100.2 (0.9) | 100.1 (0.4) | 99.9 (0.4) | 99.4 (0.4) |

**HYGROMETRY-GEOTHERMOBAROMETRY-EXPERIMENT H$_2$O**

| Parameter: | Magmatic H2O (wt.%)[b] | Magma T (°C)[c] | | | | Exp.Melt H2O (wt.%)[d] | |
|---|---|---|---|---|---|---|---|
| | PLG-LIQ | CPX-LIQ anhydrous | CPX-LIQ hydrous | OL-LIQ anhydrous | OL-LIQ hydrous | Min. | Max. |
| | ~0.8 | ~1160–1200 | ~1150–1190 | ~1160–1184 | ~1100–1144 | 0.005 | 0.02 |

**VISCOSITY**

| | | | | | | | |
|---|---|---|---|---|---|---|---|
| Experiment temperature | 1060 °C | 1110 °C | 1125 °C | 1150 °C | 1200 °C | 1250 °C | 1300 °C |
| Effective viscosity (Log10Pa s)[e] | 4.9–5.0 | 4.7–4.8 | 2.6–3.9 | 2.2 | 1.74 | 1.1 | 0.83 |
| Accuracy (+/− Pa s)[f] | 500 | 500 | 500 | 13 | 5 | 5 | 3 |
| Crystallinity (vol. %)[g] | N.D. | 34.3 | 8.5 | 4.3 | 1.4 | <0.1 | 0 |

[a]Determined by EPMA. Values in parenthesis are 1 sigma s.d. about the mean value.
[b]Based on plagioclase-liquid (PLG-LIQ) hygrometry (ref. [35]).
[c]Based on clinopyroxene-liquid (CPX-LIQ) and olivine-liquid (OL-LIQ) geothermobarometry (refs. [32, 36]).
[d]Based on FTIR measurements.
[e]Rheometer-based measurements; range of values provided for 1060, 1100, and 1125 °C runs due to time-dependent crystallization affecting measurement stability.
[f]Accuracy determined by factory-calibrated algorithm based on the instrument's absolute torque and variance thereof.
[g]Determined on BSE images with ImageJ software.

previously published geochemical and petrological analyses on Cumbre Vieja's 2021 lava[18]. Normative calculations show that this basanite is furthermore nepheline-normative (~4.8 wt.% nepheline), which in turn places it in the family of critically $SiO_2$-undersaturated magmas that differentiate with neutral, or only slight $SiO_2$ enrichments as the system crystallizes[13]. Importantly, magmas of this alkaline series do not saturate in $SiO_2$ as they crystallize and differentiate, and therefore their matrix melts will not precipitate quartz or other silica polymorphs. Hypersthene-bearing basanites, and rocks of tholeiitic affinities, by contrast, may evolve chemically towards higher degrees of $SiO_2$ enrichment and even saturation in this component as their crystallization progresses[13]. As the bulk viscosity of partially crystalline magma depends both on the matrix melt-viscosity—itself a strong function of $SiO_2$ content—and the inclusion of crystalline solids[5,31], the alkaline composition and attendant differentiation trend of this basanite should act to suppress otherwise very large viscosity increases due to enrichment of $SiO_2$.

The ash contains a small (~5–10 vol. %) complement of microphenocrysts (~40–500 μm) that comprise, in order of decreasing abundance, clinopyroxene, olivine, plagioclase, and titanomagnetite (Fig. 5). The groundmass glass also contains between 5 and 10 vol.% microlites, themselves consisting of the same phases as the microphenocryst assemblage yet being modally dominated by plagioclase. Clinopyroxene and olivine microphenocrysts also contain abundant glass inclusions, the compositions of which resemble the bulk tephra compositions (Table 1). Like the bulk rock, these glass inclusions plot within the basanite field of a standard total alkali-oxide versus silica plot (Fig. 4). The composition of matrix glasses span a range from those that mimic the bulk rock or are close to it, to slightly $SiO_2$ enriched compositions (see Tephra 1–3; Fig. 4; Table 1); these latter glasses, in addition, do exhibit significant differences in $K_2O$, CaO, $Al_2O_3$, FeO, and $SiO_2$ (~44–46 wt.%; Table 1). We attribute these differences to natural variability in the erupted magma composition, and the subtle influence of crystallization on residual matrix glass composition.

The dominance of clinopyroxene in the microphenocryst assemblage and the abundance of fresh glass in direct contact with those crystals permits the application of mineral-melt

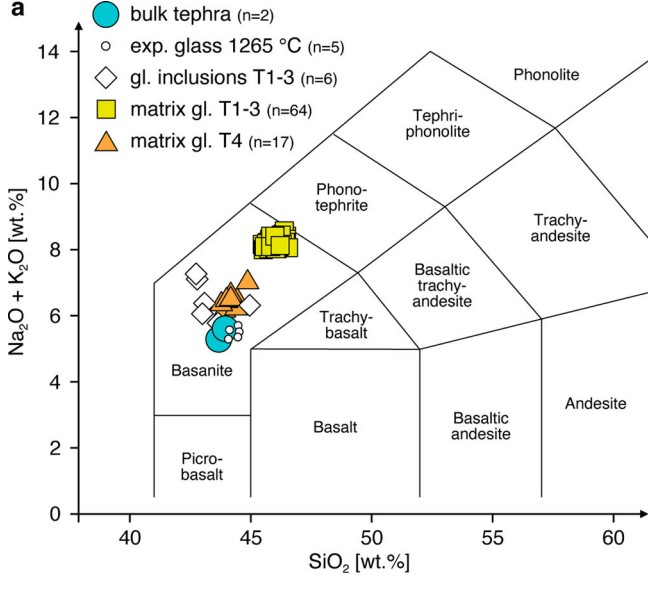

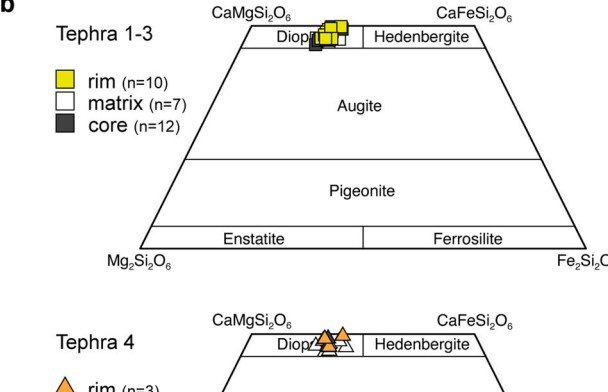

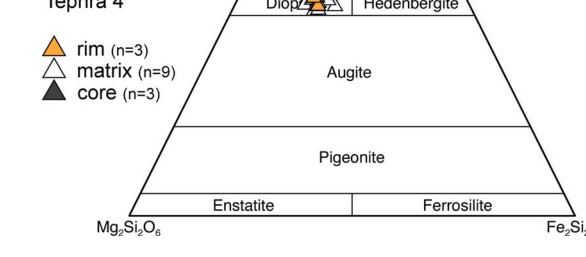

**Fig. 4 Geochemistry of 2021 Cumbre Vieja eruption products. a** Total alkali-oxides versus silica plot for a variety of natural and experimental materials. The bulk compositional information for bulk ash samples (large blue dots) along with other products indicate basanitic bulk rock composition. Glass inclusions, and some matrix glasses do not deviate significantly from the bulk rock. Glass analyses of 1-atm crystallization experiments (small open circles), which derive from the bulk ash, plot on the bulk rock and have equivalent provenance to the rheometry glasses, indicating very little compositional shift during viscosity measurements. Matrix glass in pyroclasts from tephra sample T1-3 demonstrates natural variability due to groundmass crystallization. **b** Pyroxene quadrilateral compositions of clinopyroxene microphenocrysts (filled symbols) and groundmass microlites (open symbols) in tephra collected in El Paso (Tephra 1–3) and Santa Cruz (Tephra 4). These data form the basis for geothermobarometry calculations, which yield approximate magma temperature and pressures according to refs. [32] and [33].

geothermobarometers to estimate the magma's pre-eruptive temperature and pressure. Accordingly we applied the clinopyroxene-liquid geothermometer of ref. [32] and the barometer of ref. [33] to cpx (microphenocryst rim and microlite) and matrix glass compositions (Table 1; Supplementary Information) to arrive at a relatively narrow P-T range of 1160–1170 °C and

pressure from about 7 to 10 kbar. Such values are similar to published estimates on other historic mafic eruptions at Cumbre Vieja[23,27,34]. When clinopyroxene core and glass inclusion data are used—the latter of which is taken as a proxy for the bulk magma composition—then T and P are ~1200 °C and ~9 kbar, respectively. Melt-$H_2O$ contents determined with the plagioclase-liquid hygrometer of ref. [35] using matrix plagioclase microlites and coexisting glass compositions are ~0.8 wt.% $H_2O$ (Supplementary Information). Under these hydrous conditions, T estimates shift to slightly lower values (~1150–1160 °C for rim-glass; ~1190 °C for core-glass inclusions); pressure remains relatively unchanged under the hydrous magma scenario (~7–9 kbar).

Application of olivine - liquid geothermometers[32,36] (e.g., eq. 22 in ref. [32]) to the tephra samples yields a broad temperature range (~1100–1185 °C) depending on the chosen pressures (7–10 kbar) and $H_2O$ contents (0–0.8 wt.%; "Methods"). However, owing to large (>30%) discrepancies between measured and predicted olivine-melt partition coefficients, $K_D(Fe–Mg)^{ol-liq}$, equilibrium conditions between olivine and coexisting melt were not likely met. Therefore, we consider the temperatures (~1150–1200 °C) calculated by clinopyroxene-liquid thermometry to be more robust and use these values in subsequent assessments of permissible viscosities deriving from rheometry experiments. In summary, the P-T estimates indicate a magma stored at pressures up to 10 kbar with temperatures in the range of about 1150–1200 °C.

**Viscosity measurements on Cumbre Vieja basanite.** Experimental rheometry results are presented as a temperature versus $\log_{10}$ melt viscosity ($\mu$; in Pascal seconds; Pa s) plot in Fig. 6 and in Table 1 where measurement variance and accuracy are also given. All melt $\mu$ data are considered anhydrous owing to efficient outgassing of the melts at 1-atm pressure and being stirred. The data exhibit a logarithmic increase in $\mu$ with falling T in the temperature range of 1300–1125 °C. These data also trace an Arrhenian relation when plotted in an inverse-T-log-$\mu$ space (Supplementary Fig. 1). The data can be viewed as comprising two distinct trends (Fig. 6). Firstly, a high-T trend corresponds to $\mu$'s ranging from less than 1 log unit (1300 °C) to approximately 2.5 log units (1125 °C). A second lower-T trend, from 1125 to 1110 °C, demarcates a sharp increase in $\mu$ that nearly exceeded the mechanical limits of the rheometer ($\mu_{max} \sim 10^6$ Pa s). We interpret the dramatic increase in viscosity as the result of severe crystallization[5] at T's < 1125 °C. Indeed, short duration (15–30 min), 1-atm heating experiments in which small aliquots of the basanite ash were held at a range of temperatures similar to those of the rheological measurements indicate that the basanite contains ~34 vol.% crystals after just 30 min of dwelling at 1123 °C (Fig. 6; "Methods"; Supplementary Information). These 1-atm experiments furthermore indicate relatively subtle (<0.1 to ~4 vol.%) crystallinities in melts at higher T's (~1163–1265 °C). However, we consider these crystal contents to reflect minimum values given that the short durations of these experiments (10's of mins) may not have fostered attainment of textural equilibrium at a given T.

To summarize, the two distinct viscosity traces (Fig. 6) indicate a range of effective viscosity values that are representative of high-T magma that ranges from practically pure melt (T > 1265 °C) to containing unknown but likely dilute amounts of crystals, and another low-T magma that owing to its high crystallinity, had likely developed a significant yield strength and ultimately an interlocking crystal network leading to high effective viscosities[5,37]. As a primary goal of this study is to establish baseline values of the viscosity of hot, newly emergent Cumbre Vieja magma, we will

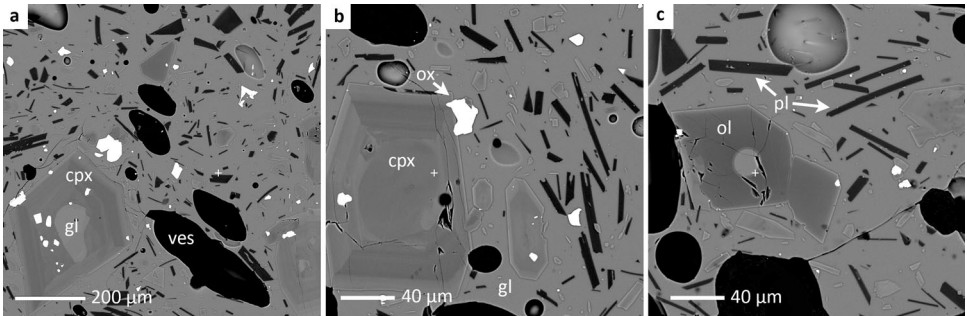

**Fig. 5 Mineralogy and microtextures of Cumbre Vieja basanite ash. a–c** Representative BSE photomicrographs of basanitic ash collected on 18 November, 2021. These images depict the dominant and widespread mineral assemblage occurring as microphenocrysts and microlites: clinopyroxene (cpx), olivine (ol), plagioclase (pl), and Fe–Ti oxides (titanomagnetite; ox). All silicate and oxide minerals were identified by EPMA analysis. Groundmass glass (gl) is abundant both in the matrix and as crystal-hosted inclusions (frames **a** and **c**), and its composition varies subtly with the amount of microlites. Glass inclusions resemble the major element composition of the bulk rock (Fig. 4; Table 1). The compositions of coexisting clinopyroxene, plagioclase microlites, and glass in the assemblage indicates pre-eruptive P and T of ~7–10 kbar and 1150–1200 °C, respectively, in addition to relatively low melt-$H_2O$ contents (~0.8 wt.%; Supplementary Information).

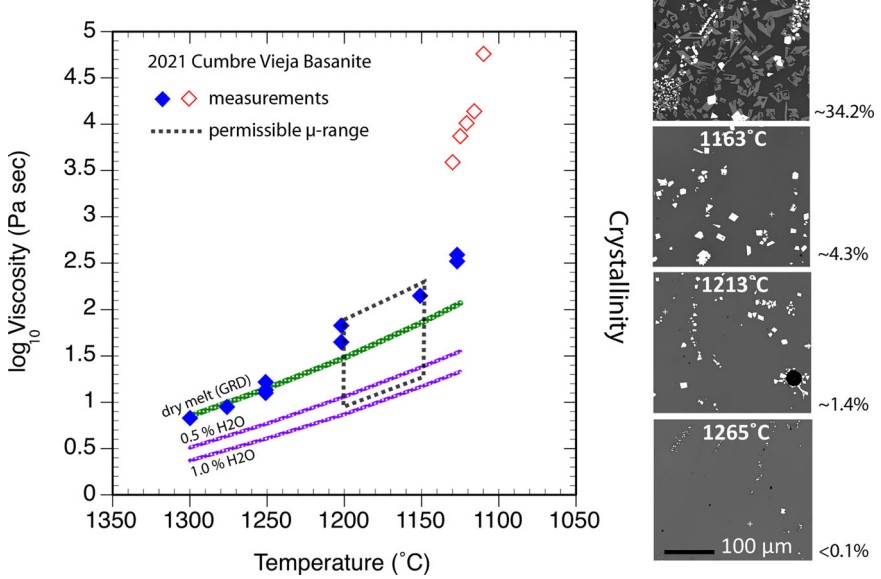

**Fig. 6 The viscosity of the 2021 Cumbre Vieja basanite.** Results of rheological measurements ($\log_{10}\mu$) performed at 1-atm and temperatures from 1300 to 1110 °C. Owing to the 1-atm conditions of these stirring experiments, the melts became thoroughly degassed and thus, the data (blue diamonds) represent nominally anhydrous viscosities. All viscosities are *effective values* as the melt contained crystals throughout the experiments; BSE images on the right demonstrate crystal contents (determined from BSE image analysis) as a function of temperature in separate 1-atm heating experiments. The smooth shift towards higher viscosities with falling temperature interval from 1300 to 1125 °C (blue diamonds) demarcates viscosities of low-crystallinity melts. A step-function change to higher viscosities occurs at T < 1125 °C (open red diamonds) and highlights the influence of high crystal contents (e.g., as shown in the top right BSE image). Predicted basanite viscosities derived from the GRD viscosity model[7] are indicated by the green and two purple curves, the latter of which track hydrous melt viscosities for a basanite melt containing 0.5 and 1.0 wt.% $H_2O$. As the 2021 basanite may have contained up to 0.8 wt.% $H_2O$, its *minimum* viscosities will plot between the purple hydrous GRD curves within the permissible eruption temperature range of 1150–1200 °C. In total, Cumbre Vieja's effective eruptive viscosity range can be bracketed by the area enclosed by the bold dashed line; this range accounts for all the likely parameters including T, composition, crystallinity and hydrous components in the melt.

focus on rheological values that derive from the high-T segment, and in particular, those that most closely overlap with possible eruption temperatures (1150–1200 °C; Table 1).

The experimental $\mu$ data, ranging from less than 0.9 log units (~8 Pa s) at 1300 °C to slightly more than 2.5 log units (~330 Pa s) at 1125 °C, show slight curvature within of the high-T trend (Fig. 6). This observation could reflect subtle non-Arrhenian behavior as predicted by multi-component silicate melt viscosity models[7], or, could mark the influence of crystals within the melt (see discussion in the following paragraph). Most importantly, the

viscosities in our study compare well with values determined in previous experimental studies on similarly composed basanites[4], in addition to predictions of the widely applied multicomponent viscosity model of ref. [7] (hereafter denoted GRD), which, as depicted in Fig. 6, are based on the magma's XRF-determined bulk composition (Table 1). This all suggests that our measurements are both accurate and illustrative of maximum anhydrous viscosity values.

Figure 6 provides a direct comparison between our experimental $\mu$'s and GRD-predicted $\mu$'s, and demonstrates a one-to-

one correspondence of the respective values in the range of 1300–1250 °C. At $T \leq 1250$ °C, experimental $\mu$ deviates upward from the GRD-model, most noticeably in the range of 1200–1125 °C, which could reflect the presence of crystals in the experimental melt, and the tendency of crystals to increase the bulk stress and therefore the effective viscosity of the magmatic suspension[37–39]. We have investigated the feasibility of this interpretation by calculating the relative viscosity ($\mu_{rel}$)[31,37]—the viscosity of the crystal-melt suspension normalized by the pure-melt viscosity ($\mu_{rel} = \mu_{sus}/\mu_{melt}$)—of a dry basanite magma containing modest (~6 to 16 vol. %) amounts of crystals (Supplementary Information). These crystal contents are consistent with what is observed in the tephra samples (Fig. 5; Supplementary Information), and are used as a guide to investigate the potential effects of crystallinity on rheology. The crystal contents, along with crystal size and shape distributions needed to perform the rheological calculations (via ref. [31]), derive from textural measurements that we made on six tephra clasts sub-sampled from the bulk ash that also served as experimental starting material (Supplementary Information). These calculations show that the effects of the tephra's crystal cargo—taken as a proxy for what could have conceivably been present in the freshly erupted lavas—on the viscosity of the suspension is small, as reflected by relative viscosities of only ~1.3 to 1.9 (Supplementary Information). These relative viscosities mean that all crystal-free estimates of melt viscosity, including the estimates for anhydrous basanite derived from the GRD model (Fig. 6), could profitably be adjusted up, ie., multiplied by the relative viscosity to derive possible effective crystal-bearing melt viscosities. Thus applying these corrections to the GRD model viscosities in the experimental range of 1200–1150 °C (Fig. 6) will close the gap between experimental viscosities and the adjusted GRD values. In other words, the offset between measured and predicted anhydrous melt viscosities observed in Fig. 6 are likely due to the presence of a small quantity of crystals in the experimental melts.

Crystals may also push a suspension into the non-Newtonian realm—for example shear thinning behavior—due to crystal-crystal interactions and consequent shear heating of the suspending melt[31,40,41]. This would be manifested as unstable or oscillating $\mu$, and failure to reach a steady state in rheology experiments. These phenomena only occurred in our experiments at low T ($< 1125$ °C), in which the melts were crystal-rich and showed very high effective $\mu$ (Fig. 6).

Of the viscosity data, the most relevant to understanding the rheological state of near-vent, freshly emergent basanite magma, are those falling within the estimated eruption temperature range (~1150–1200 °C; Table 1). Thus, effective anhydrous magma viscosities are about 50 Pa s to a little over 160 Pa s. These values, while significantly lower than recent tholeiitic lava eruptions (~250–1150 Pa s)[2,42], are quite comparable with field-based $\mu$ estimates of historical Hawaiian lavas that exhibited exceptionally fluid, near-vent behavior, including supercritical flow phenomena[8,43] (~$10^2$ Pa s). In the high likelihood that the Cumbre Vieja basanite was even slightly hydrous—conditions that are supported by the explosive nature of the eruption, plagioclase-glass hygrometry[35] (Table 1; Supplementary Information), the bulk basanite tephra's loss-on-ignition (L.O.I.; Table 1), and previous findings on other Cumbre Vieja basanites suggesting magmatic $H_2O$ contents of ~0.5–1.0 wt.%[23,34,44]—then both pure- and crystal-bearing melt viscosities would be substantially lower. Calculations using the GRD model (Fig. 6), indeed indicate that hydrous basanite melt (crystal-free) containing ~1.0 wt.% will be more than an order of magnitude less viscous than the anhydrous melt from 1150 to 1200 °C (e.g., as shown by hydrous melt $\mu$ GRD curves in Fig. 6; ~15 Pa s to ~7 Pa s). With

~0.5 wt.% $H_2O$ the viscosity is only slightly higher than the 1.0 wt.% melt, varying from about 23 to ~11 Pa s from 1150 to 1200 °C. These GRD-determined values bracket what the permissible crystal-free melt viscosity of the 2021 Cumbre Vieja magma would be: basanite melt with an estimated 0.8 wt.% $H_2O$ is just ~18 to 9 Pa s across the eruptive temperature range. In the likely case that a small complement of crystals were present in the hydrous melt (e.g., Figs. 5, 6), the effective magma viscosities[31,37] would be slightly higher (ie., by a factor of 1.3–1.9) than the pure-melt values, but ultimately would not exceed a few tens of Pa s. Such ultralow viscosities are not typical of basaltic magma[4,9], and instead, resemble those of the ultramafic komatiite melts[45] and nephelinite lavas[17].

In summary, the viscosities measured across a range of eruption temperatures incorporate the effects of melt composition and crystallinity and thus can be viewed as effective anhydrous values. In the natural system, the magma likely carried some $H_2O$ in solution, which in turn means that in nature, Cumbre Vieja basanite erupted with ultralow viscosity, thus dictating a myriad of observed magma transport and eruptive processes.

## Discussion

Several aspects of the near-vent activity at Cumbre Vieja signal special and rarely seen physical transport properties (e.g., low-viscosity and supercritical flow phenomena), that influenced the eruption behavior of this magma (Figs. 2, 3). These characteristics establish a clear link between fundamental magma parameters (e.g., temperature and composition), transport properties (viscosity), and eruption behavior, which was ultimately responsible for the pervasive destruction this event caused. The main phenomenological observation is that the lava was exceptionally fluid. Because of this, lava was able to translate great distances (km's in just hours to days; Fig. 1) before significantly cooling and succumbing to the rheological barriers built by crystallization. Even after some cooling and crystallization occurred, however, the lava still possessed a low apparent viscosity due to the low silica content of the matrix melt (Table 1; Fig. 4). Our experiments bear out these observations (e.g., they span a range from mostly molten to slightly crystalline) and provide an explanation for the high rates of travel, the lavas' odd hydrodynamic behavior, including the high exit and outflow velocities (Fig. 2) and the formation of standing waves (Fig. 3).

In order to gain further insights into how the basanite's low viscosity may have influenced flow parameters that underpin near-vent eruptive behavior—and by extension transport in the shallow sub-surface—we must assess dimensionless parameters that describe flow regimes. The Reynolds number ($Re$), which is a measure of the relative importance of inertial to viscous forces, is a sensitive function of viscosity and, given the orders of magnitude range of natural magma viscosity[7], provides a convenient way to define the onset of turbulence in fluids[20]. We can estimate $Re$'s of the near-vent lava cascades (Fig. 2) with the following form[8,21]:

$$Re = \frac{\rho v D}{\mu} \qquad (1)$$

Here, $\rho$ is density (kg m$^{-3}$), $v$ is velocity (m s$^{-1}$), $D$ is the hydraulic diameter, or characteristic length (m) defined for open rectangular channels as four times the product of the channel depth and width divided by the wetted perimeter ($P = 2h + w$)[8,21,46]. As we have few constraints on the cross-sectional geometry of the cascade channels (Fig. 2), and owing to the fact that these proximal flows did not appear to enter pre-existing, mature lava structures, we will operate on the

assumption that they are relatively thin, tabular flows akin to sheets moving down channels of shallow rectangular cross section[8]. The viscosity, $\mu$, is the apparent (or effective) viscosity, in Pa s. As we are utilizing dynamic information gained from remote observations, the flow dimensions (channel width and depth) carry some error; we estimate uncertainties of ±2 m in channel width and ±0.2 m in the depth values (Supplementary Information). Due to the irregularity of the channels themselves (Fig. 2), flow depth and width are not likely constant along the lengths of the cascades. We therefore consider $Re$ estimates to be accurate to an order of magnitude. Using an approximate, and minimum 0.3 m flow depth, a width of 3 m, magma density of 2700 kg m$^{-3}$ (ref. [47]), a velocity of 10 m s$^{-1}$ (determined from video analysis of the lava cascades; Fig. 2), and $\mu$ of 20 Pa s—an estimate based on a hydrous (~0.8 wt.% H$_2$O) basanite at an intermediate eruption T of 1175 °C (Fig. 6) and adjusted for the effect of crystals by way of a feasible relative viscosity of 1.6—yields $Re$ of ~1350. Clearly, $Re$ will be higher for a lower melt viscosity, e.g., corresponding to a higher eruption temperature (~1200 °C; Table 1). Additionally, $Re$ will rise considerably for greater characteristic lengths (e.g., flow depth) and velocities. A potential upper limit in $Re$ of ~1930 is reached by applying an effective viscosity of 14 Pa s, which is the minimum permissible hydrous (0.8 wt.% H$_2$O) crystal-bearing basanite viscosity (Fig. 6; Table 1), and using all other previously defined variables.

The effect of reduced bulk magma density, as is caused by the presence of bubbles, would lower $Re$ estimates by an amount equal to the fractional bubble concentration. The reduced density effect of bubbly magma would however be readily offset by the flows having greater thickness. For example a flow of 30% porosity but of a marginally higher 0.5 m thickness (in contrast to 0.3 m considered earlier) will have a $Re$ in excess of 2200. We have furthermore not considered the potential effects of bubbles on the effective viscosity due to a lack of constraints on lava vesicularity during eruptions of the cascade lava. At high flow shear strain rates, bubbles would be highly deformed, leading to a high capillary number regime and consequently negligible effects on the effective viscosity[2,6].

The $Re$'s estimated here exceed previous $Re$ estimates for basaltic lavas by a factor of two to more than an order of magnitude[8,20,21], yet overlap with the lower limit of $Re$'s of komatiites[46,48] (~$10^3$–$10^6$), lavas of which are thought to have been emplaced in the turbulent flow regime[45]. Thus, $Re$'s indicate that some Cumbre Vieja basanites erupted within and perhaps just beyond the transitional regime between laminar and turbulent flow (ie., $500 < Re < 2000$)[8,9,49]. In this flow regime, lava behaves in a disrupted state, characterized by episodes of laminar flow giving way to intermittent bursts of turbulence[9,49,50]. Thus rheological and $Re$ analysis suggests that some of the Cumbre Vieja basanite (ie., that observed on 18 November 2021) emerged in a highly fluid, possibly turbulent state[9,50].

High $Re$'s and ultralow viscosities are consistent with the observation that lava in the cascades (Fig. 2) appeared to propagate as a series of fronts that would wet the channel behind them giving way to several subsequent lava pulses that, instead of coalescing into a continuous flow, would maintain their blunt forms and flow independently of both earlier and later pulses. This behavior is qualitatively similar to flow phenomena exhibited by turbulent water flows, including bores, foreshore swash waves, and the propagation of water drops down wetted surfaces[51,52]. In all of these cases, spatially distinct flow discontinuities are characteristic features indicative of the formation of hydraulic jumps[51]. The lava cascade flow fronts (Fig. 2), interestingly, remained bright as they progressed downslope, showing that crust formation was inhibited. This in turn could reflect turbulent mixing resulting from the conversion of

momentum to turbulent eddies and circulation in these flow fronts, not unlike in the head of a density current[53]. The tops and trailing parts of the flows, by contrast, attained a slightly darker appearance, perhaps reflecting the onset of thin-crust formation. The video evidence shows however, that these dark zones do not persist for long (~seconds) before they become re-incorporated into the body of the flow (Fig. 2). Confirmation of the dynamic similarity between low-viscosity water systems and the Cumbre Vieja lava will, however, require further analysis, as our remote observations are limited and do not provide critical cross-sectional shape information to draw further comparisons.

High $Re$ lava flows also exhibited macroscopic surface undulations that reflect the growth and propagation of gravity waves[20]. Such features, commonly termed standing waves[21], were present in near-vent lava streams at Cumbre Vieja (Fig. 3) and indicate flow in a supercritical state, which furthermore is characterized by large Froude numbers[20]. The Froude number ($Fr$), which measures the relative importance of inertial force on a fluid to its weight can provide an indication of the dynamical regime under which the flow occurs[20]:

$$Fr = \frac{v}{\sqrt{gh}} \qquad (2)$$

Here $v$ is the velocity (m s$^{-1}$), $g$ is the gravity constant (9.81 m s$^{-2}$), and $h$ is the flow depth (m). Video analysis of the flow depicted in Fig. 3 (Supplementary Information), suggests an approximate velocity of 7 m s$^{-1}$. If this flow were 1 m thick, $Fr$ would be about 2.2. Clearly great uncertainties exist in this flow's dimensions and therefore velocity estimates. Nonetheless, uniformly high $Fr$ numbers (1.6–3.2) are implied for a range of reasonable velocities (~5–10 m s$^{-1}$). Generally, for $Fr > 1$, the flow will occur in a supercritical regime characterized by hydraulic jumps[20,21] and the persistence of gravity waves whose relative speeds are much less than the overall flow velocity.

In summary, the observations of near-vent flow phenomena on 18 November 2021 in addition to the occurrence of standing waves that formed days later (Fig. 3) validate the experimental evidence for ultralow viscosity of the basanite, the high calculated $Re$'s, and its consequent emplacement as rapidly moving supercritical lava flows[20]. To our knowledge, the high $Re$ and $Fr$ assessed here are relatively uncommon among rheological investigations of basaltic lavas[16,21], and underscore how the low viscosities of this basanite fostered its rarely observed and rapid eruption behavior. Indeed, the rheological measurements suggest that the 2021 Cumbre Vieja basanite is among the least viscous basaltic magmas observed on Earth (Fig. 6). The 1-atm, anhydrous state of the basanite and associated rheological measurements establish a baseline that demarcates the maximum viscosity across the range of possible eruption temperatures (~50–160 Pa s). However, in the likely case that magmatic water was present in the basanite, effective viscosities would have been substantially less, in the range of about 10 to a few 10's of Pa s. These ultralow viscosities place the Cumbre Vieja basanite into a distinct behavioral class from typical basalts, in that some of its near-vent flows were characterized by very high $Re$ and $Fr$ numbers. The physical implications of these conditions include flows that were both turbulent and supercritical in nature. These rare transport characteristics could have fostered not only the magma's rapid conveyance from the mantle, but also the relatively unhindered transport of magma in the shallow sub-surface, to both the long-lived cone and the seemingly spontaneously formed new vents throughout the eruption.

Typically we associate highly hazardous activity with high-Si explosive volcanism[54]; however, the eruption of Cumbre Vieja has shown that long-lived effusive eruptions of highly fluid magma may be just as destructive and equally difficult to forecast

and respond to. Ultralow magma viscosities have only rarely been estimated in mafic eruptions[8,9,12], in turn suggesting that many physical eruptive processes may entail and require a different hazards response. The recent (2018) flank eruption of Kīlauea, for example, produced relatively hot (~1140 °C) and fluid magma from the third phase of its activity[55]. However, viscosity estimates for this phase[2,42] (250–1150 Pa s) are still at least an order of magnitude higher than the effective viscosities of the 2021 Cumbre Vieja basanite.

## Methods

**General Information**. Additional experimental, analytical and geochemical information relevant to these methods and the presentation of data in the main body text are contained in an expanded supplementary information section. The following sections provide the salient methodological details for this study.

**Sampling**. The samples investigated comprise fine to coarse ash from air fall tephra deposits sampled in both proximal (~3–4 km distant) and distal (~12 km distant) locations with respect to the 2021 El Paso (Cumbre Vieja, La Palma) vents (Fig. 1). These materials were collected during the week of 14–20 November 2021 from flat surfaces that include a hotel's roof (in Santa Cruz), the field vehicle, and cardboard sheets set out to capture the ash as it fell (in El Paso). Due to dry weather, the samples have not undergone any post deposition alteration by surface processes and were swept into airtight plastic bags shortly after deposition.

**Viscosity measurements**. Rheological experiments were performed using a Brookfield DV3T rotating cylinder rheometer coupled to a Thermoconcept vertical elevator tube furnace. First, ash was introduced into a Pt crucible (350 ml) and subsequently melted at 1200 °C. Ash was then repeatedly added to the crucible until the melt reached a level to within 4 mm of the surface of the crucible. Then, the rheometer spindle was inserted to the melt through an opening in the top of the furnace, while taking care that the insertion depth was the same for all measurements (ca. 2 mm above the base of the crucible). Viscosity measurements were performed at different temperatures (1110–1300 °C) for durations typically on the order of several hours, until the viscosity reading reached a plateau which was interpreted to reflect that the melt attained its crystal-melt equilibrium at the corresponding temperature.

Two cylindrical spindle end pieces were used to perform viscometry, each being made of Pt–Rh (800/200 alloy) for its temperature stability and chemical inertness. The spindles comprised different diameters (3 mm vs. 9 mm) to expand the range of possible melt viscosities for given rheometer settings. The thin spindle was used to investigate viscosities in the range of approximately 1000–10,0000 Pa s, and the larger spindle to measure viscosities ranging from less than 10 to about 100 Pa s.

Experimental temperatures were monitored with a furnace internal thermocouple and one external K-type thermocouple that was introduced through the furnace's roof. The external TC was lowered to both the surface of the melt and inserted to the melt base via a platinum sleeve to document any potential gradients across the sample. Each TC measured with good precision (±3 °C), however, a temperature offset was observed amounting to 50–60 °C between the external TC (reading higher) and the furnace TC. This offset is attributed to a thermal gradient arising from the opening at the top of the furnace where the spindle is introduced. This temperature difference persisted across the entire range investigated and thus all reported temperatures in this work are the external TC temperature.

The instrumental error on viscosity measurements is minimized for torque values in the range of 10–90%. Thus, for all measurements a spindle size and rotation rate were chosen to achieve sufficient torque on the sample/spindle arrangement (typically 10–20% torque). The rheometer's measurement precision is indicated in measurement reports as ±10–230 Pa s. Thus, the higher end of this range is considered to be the maximum error on our measurements. The accuracy of viscosity measurements was determined by calibration of the instrument with viscosity standard fluids (Cannon N270000, N1900000, and N2400000) whose individual temperature-dependent viscosities span the expected range of natural basaltic lava[20]. Calibrations were made separately for each spindle end piece. Factory-supplied fluid viscosities were reproduced by the rheometer to within ≤10% error of the data sheet values. Calibration fluid measurements were repeated 3–5 times before and after unknown measurements of high-$T$ experiments to check for instrument drift which revealed no significant deviations.

**Melt-crystallinity determinations**. Given the temperature range investigated, crystals likely nucleated and grew at some point during the experiments. Thus, the measurements could reflect either pure-melt (liquid only) values (e.g., at high-T) or bulk viscosities that mark a suspension rheology (e.g., at low-T). In order to assess whether the melt was crystal-bearing at the different experimental temperatures, crystallinities were investigated in a rapid spindle-pull experiment and in a separate experimental series that is described in more detail below and in the Supplementary Information.

A mostly crystal-free rheology at the highest temperatures was confirmed by pulling the spindle from the melt residing at ~1200 °C and quenching the adhered melt in water. Subsequent microscopic evaluation of these glasses revealed only a small fraction (<1 vol.%) of small (<50 μm diameter) sparse anhedral oxide minerals that were interpreted to have been undergoing dissolution at the time they were quenched. However, at temperatures less than 1200 °C, higher crystallinities were observed.

In order to constrain the crystallinities of experiments as a function of temperature, and to assess the potential related rheological influence of those crystals on viscosity measurements, a separate series of crystallization experiments in the range of 1265–1123 °C was carried out. Crystallinity experiments were performed by loading small quantities (~0.1 g) of the starting basanite ash into five, 5 mm diameter Pt-cups, heating these in a Nabertherm muffle furnace, and quenching them subsequently in a water bath. The temperature was monitored with both an internal TC and external K-type TC, which was positioned directly next to the Pt cups (~4 cm) and the offset between the two TCs was about 13 °C. The furnace was heated to 1265 °C over 1 h and let dwell at this temperature for 15 min after which one crucible was removed with Pt-tongs and quenched. The procedure of dwell and following quench was repeated for subsequently lower temperature steps of 1213, 1163, 1138, and 1123 °C. These were instigated automatically over a duration of ~5 min by the furnace's temperature control unit. Dwell periods of 15 min were employed for all but the lowest two temperatures, for which 30 min were used to enable the melt to approach equilibrium crystallinity. The quenched melt-filled crucibles were embedded in epoxy, cut in half along the axis of the cups, and polished for inspection on the EPMA. Backscattered images of the different samples were then investigated for crystal content using ImageJ software.

*Textural measurements and effective viscosity calculations*. We calculated the effect of crystals on the viscosity of the Cumbre Vieja melt by incorporating textural measurements into a suspension rheology model[31] that outputs relative viscosity. Textural measurements were made on six representative tephra clasts with Image J software, and included the crystallinity, and crystal size and shape distributions of all important mineral phases (ie., olivine, plagioclase, clinopyroxene, Fe–Ti oxides). Textural parameters were in turn used to calculate the maximum crystal packing fraction and crystal population polydispersity via the formulation of ref. [31]. Our specific approach and the results of these textural measurements are described further in the Supplementary Information, including an example of the textural measurement steps.

**Geochemical and Analytical Techniques**. Bulk rock analyses of tephra collected in November 2021 were conducted by XRF analysis (Table 1) while in situ *c*hemical analyses of natural samples and products of the crystallinity experiments were carried out using a JEOL JXA 8200 electron microprobe (EPMA). EPMA analysis protocols of glass, Fe–Ti oxide, and silicate phase analyses included an acceleration voltage of 15 kV and a beam current of 12 nA. Glasses were analyzed for all major elements and additional F, Cl, and $SO_3$ using a beam diameter of 10 and 5 μm for glass inclusions. Silicate phases were analyzed for major elements using a beam diameter of 5 μm. Survey measurements on glass were carried out to assess Na-drift but revealed no significant Na-loss. Reference materials VG-2, VG-A99, and a natural obsidian standard were analyzed repeatedly during each session analyzing glass, while diopside, wollastonite and orthoclase standards were used for silicate phase measurements. Matrix correction was carried out for glass and silicate phase analyses using the ZAF and PRZ methods, respectively.

**Geothermobarometry and hygrometry**. Approximate magma/eruption temperature and hydrous component values were determined by application of crystal-melt geothermometers and hygrometer. The glass compositions of each sample was averaged from the microprobe analyses and employed with individual clinopyroxene analyses by applying equation 33 of ref. [32] in concert with the geobarometer of ref. [33]. For the calculation of eruption temperature(s) which is used to narrow down the range of viscosity estimates, analyses of clinopyroxene rims and matrix microlites were used, interpreted of being representative of the most recent melt-crystal equilibrium. The glass compositions measured by EPMA were mostly anhydrous, however, considering the eruption behavior (e.g., explosive activity and lava fountaining) the magma must have comprised at least some $H_2O$ to drive this activity. Therefore, in order to obtain an estimate of magmatic water content during the eruption of Cumbre Vieja Basanite, the improved calibration of the plagioclase-liquid hygrometer of ref. [31] was employed. Average compositions of matrix plagioclase crystals used with the hygrometer are $X_{An}$ 0.63 (Tephra 1–3) and $X_{An}$ 0.68 (Tephra 4). Both, the clinopyroxene-liquid geothermobarometer and the plagioclase-liquid hygrometer were applied iteratively to determine the magmatic water content and the hydrous clinopyroxene crystallization conditions. Analyses of clinopyroxene cores of Tephra 1–3 were additionally investigated and employed together with glass inclusion data of primitive composition of the same samples (Table 1) to calculate a possible onset temperature of crystallization.

Equilibrium between all the investigated cpx – liq composition pairs was ensured by examination of both, the Fe–Mg exchange between melt and crystals and the DiHd components (Diopside-Hedenbergite). Only cpx – liq composition pairs that were within 10% of DiHd component equilibrium and at the same time satisfying that the measured $K_D(Fe–Mg)^{cpx-liq}$ value was within ±25% of the

predicted temperature-corrected $K_D$(Fe–Mg)$^{cpx–liq}$ (eq. 35, ref. [32]) were included in the geothermobarometry and hygrometry calculations.

In addition to clinopyroxene–liquid geothermobarometry, olivine (ol)–liquid (liq) pairs were investigated to determine magma temperature by applying two different published ol–liq geothermometers[32,36]. For this purpose, individual olivine rim and core compositions were used in concert with averaged glass compositions of the different samples. For anhydrous conditions, the model of ref. [36] was applied while for hydrous conditions (0.8 wt.% $H_2O$) ref. [32] (eq. 22 therein) was used. These models yield temperatures of 1160–1184 °C (anhydrous conditions), and 1100–1144 °C (hydrous conditions), which, while also dependent on pressure (7–10 kbar), are roughly in line with temperatures estimated by cpx–liq thermometry. However, as indicated by a significant deviation (>30%) of the majority of the measured $K_D$(Fe–Mg)$^{ol}$ values obtained from olivine compositions compared to the predicted $K_D$(Fe–Mg)$^{liq}$ values calculated from the glass composition[32], it cannot be verified that the olivine crystals were in equilibrium with the coexisting melt. Disequilibrium conditions are further supported by apparent zoning as indicated by bright and resorbed rims of olivine crystals in backscattered electron images (Fig. 5c). Therefore, temperatures calculated from olivine–liquid thermometers, even though broadly in line with those obtained by clinopyroxene–liquid compositions, were not considered for the assessment of probable eruption temperatures.

**Videography and lava flow regime assessment**. Open access video from the RTVC[19] Spanish/Canary Island media collective was used as the basis for flow velocity and channel dimension estimates. The video was part of live streaming feeds that were filmed on 18 November 2021 at approximately 20:00–20:45 h and on 25 November 2021 at about 09:26 h. Details of the cameras' optical characteristics are unknown; however, both cameras filmed the activity from a distance of ~5 km southwest of the vent in the town of Tajuya. In addition, field photos and other digital videos were taken from more northerly vantage points (in the city of El Paso) which were made oriented approximately perpendicular to the direction of the RTVC cameras, thereby providing a second dimension such that the approximate scale of the lava flow path and slope could be assessed. The main RTVC video of the central cone (Fig. 2) depicts two newly formed vents on the west side and downslope from the main long-lived vents. The two bubbling vents fed two independent yet parallel flowing lava streams that descended a steep rampart for about 100 m. The length of the cascade slope was estimated using the apparent vertical drop in the camera view and our orthogonal views of the activity. The hydraulic phenomena were observed within each channel and appeared to be linked to velocity increases (accelerations) that the flows experienced as they progressed away from the sources. The features of note are the arcuate structures that periodically form within close proximity (<30 m) to the lava source. The perimeters of these structures became brighter as they moved downslope, while the interiors remained relatively darker (Fig. 2). These bright perimeters are interpreted as blunt flow fronts that reflect hydraulic jumps created as a turbulent dissipation of the flows' downslope momentum. These structures served as a useful indication of the position and progress of the flow with time and thus were used to estimate flow velocity. Because the view of the video is not perfectly orthogonal to the sloping cascade, and some uncertainty to the true length of the lava run stemming from measurement errors for slope, the velocity estimates are considered to be apparent velocities and less than the true velocities. The apex of the arcuate structure (resembling ogives on lava flows) was tracked frame-by-frame during the video analysis. The travel times were documented with a stop-watch, initiated at the first signs of brightness on the leading edge of the structures. The effective distance is less than the ~100 m fall distance because the structures typically developed after a short run-out distance from the source (Fig. 2). We chose an average run distance of about 70 m to account for this behavior. Additional details on video analysis of Cumbre Vieja's lava flows and related flow velocity determinations are given in the Supplementary Information.

## Data availability

All data generated in this study are included in the manuscript, figures, tables, and supplementary information. Raw rheometry data can be accessed by request from the authors.

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

## Acknowledgements

We are grateful for analytical support from N. Groschopf and S. Buhre, especially during electron microprobe analyses.

## Author contributions

Data collection, synthesis, interpretation, and writing of the manuscript were undertaken equally by J.M.C. and Y.F.

## Funding

## Competing interests

The authors declare no competing interests.
