## [Peer Review File · Nature Communications]

REVIEWER COMMENTS

Reviewer #1 (Remarks to the Author):

This is a study of the viscosity properties of the recent basanite eruption of Cumbre Vieja volcano, La Palma, both from field observations (standing waves) and lab experiments. Overall, I thoroughly enjoyed reading this paper and I consider to be appropriate to publish it in Nature Communication. Eruption of mafic alkaline lavas are rare, and the Cumbre Vieja eruption provide an excellent opportunity to gain insight on the rheological properties of these lavas. In my opinion, the manuscript only need minor revisions. My main comments concern the calculation of lava viscosities before the discussion section. I detail my comments line by line in a separate pdf, and hopefully the authors find them constructive to improve their paper.

Reviewer #2 (Remarks to the Author):

The manuscript titled "Eruption of ultralow-viscosity basanite magma at Cumbre Vieja, La Palma, Canary Islands" by Castro and Feisel presents measurements of the viscosity of material erupted from Cumbre Vieja in 2021, estimates of eruption temperature and melt water content from cpx-liquid thermobarometry and plag-liquid hygrometry, and observations of channelized lava flow dynamics. The authors demonstrate that the fast-moving lava flows that devastated the nearby community could be the result of ultralow viscosity lava and that this ultralow viscosity was reflected in the observed supercritical flow features. This type of work is critical for hazard assessment and mitigation, as fast-moving lava flows are a serious threat to nearby populations, thus this work is significant for the field of volcanology.

I find this work to be well-written and compelling, with just a few areas for wording clarification and typo correction. The data appear to be sound and collected in an appropriate manner. However, with regards to the interpretation of the results, I do have three general points that I think would benefit from additional consideration from the authors:

1) How representative is the ash of the material that fed the fast-moving lava flows, particularly in terms of crystal microtexture? From the description of the eruption that the authors provide, the ash is primarily sourced from vents that had a "duller, orange-incandescent" lava effusion, in contrast with the vents of interest that effused lava that was "extremely bright" and "emitted a white radiance" (lines 114-115). This question ties into my next two points.

2) If the ash is indeed representative of all material being erupted, I am not certain that the rheometry measurements capture the bulk properties of the lava. From the BSE images in Figure 5, the crystal cargo was primarily dominated by elongate plag and pyroxene crystals, up to 30 vol. %. Since the size of the ash is quite small, it can safely be assumed that the texture of the magma as it erupted was quenched, with little modification from cooling. In contrast, the BSE images of the melts quenched from the rheometer experiments in Figure 6 show primarily equant oxides. The most comparable to expected eruption temperature experiments shown are the 1213 and 1163°C experiments, but those show vastly different textures to the erupted products, in terms of overall crystal content and crystal shape. Although the crystal content in the natural products is relatively low, the dominance of high aspect ratio crystals could induce changes to viscosity at much lower volume fraction (e.g., Mueller et al., 2011, Geophysical Research Letters; Cimarelli et al., 2011, Geochemistry, Geophysics, Geosystems). I would like to see whether the increase in measured viscosity relative to the crystal-free value predicted by GRD is comparable between the experiments and the theoretical values predicted by models such as Costa et al. 2009 (Geochemistry, Geophysics, Geosystems) for elongate particles.

3) For the Re calculations in the discussion, it appears that the authors use only the pure melt viscosity when considering the hydrous melt case? The crystals present in the BSE images in

Figure 5 should still be accounted for. If their effect turns out to be negligible overall in the Re calculations, it should still be mentioned.

Overall, these points are meant to clarify whether the lava is truly “ultralow viscosity” in terms of bulk viscosity, though certainly the pure melt is quite low in viscosity. It does not seem like ultralow viscosity is a prerequisite for the observed standing waves as they seem to be quite common in Kīlauea eruptions when effusion rate is high (Le Moigne et al., 2020). I do find it hard to evaluate the necessity of ultralow viscosity for the other observed flow abnormality (i.e., the propagating wave fronts in the cascades) as I am unaware of any studies noting their presence in other eruptions. Even the rapid flow emplacement times are not a “smoking gun” for an ultralow bulk viscosity lava because the emplacement times the authors list for this eruption are comparable to the emplacement times for flows from the later stages of the Kīlauea 2018 eruption. The flows from fissures 20/22 traveled 5.5 km in ~28 hours (deGraffenried et al., 2021, Bulletin of Volcanology) and the major flow from Ahu`ailā`au (formerly fissure 8) traveled 13 km in 6 days (Dietterich et al., 2021, Bulletin of Volcanology). Many of these factors tie into effusion rate, as well as bulk viscosity, so it is important to ensure which parameter is the cause.

Line by line comments:

Lines 17-19: if my second general point holds and the crystals do indeed impact viscosity in a non-negligible way, sentences like this one will need to be revised throughout the text as crystallinity does play a role, though not to increase viscosity by increasing melt SiO₂

Line 75: “a elongate” should be “an elongate”

Line 140: “strea” should be “stream”

Line 142: “Kilauea” should be “Kīlauea”

Line 173: should be “These results”?

Line 195: missing the opening parentheses before Fig. 4 call

Line 263: Saying that crystals increase “bulk stress” is somewhat confusing to me, as stress is a force applied to the system. Crystals do change the response to stress, however. This sentence should be revised to be clearer what the authors mean.

Line 278: “conditions that are supported by”

Lines 280-281: Are these water values representative of water contents at depth or what is still dissolved in the glass upon eruption? Only the latter will impact lava melt viscosity, and 1 wt.% seems awfully high for what is dissolved in a melt upon eruption for a low viscosity melt that should have no trouble degassing.

Line 334: How was lava flow depth assessed?

General grammar comment: the authors switch between using an Oxford comma and not – either one is fine, but the authors should be consistent in the style they choose.

Table 1: it would be useful to list the starting crystal contents of the tephra as well

If any of my points are unclear, the authors are welcome to contact me for clarification!

Sincerely,
Rebecca deGraffenried

Reviewer #1 PDF

This is a study of the viscosity properties of the recent basanite eruption of Cumbre Vieja volcano, La Palma, both from field observations (standing waves) and lab experiments. Overall, I thoroughly enjoyed reading this paper and I consider to be appropriate to publish it in Nature Communication. Eruption of mafic alkaline lavas are rare, and the Cumbre Vieja eruption provide an excellent opportunity to gain insight on the rheological properties of these lavas. In my opinion, the manuscript only need moderate revisions. My main comments concern the calculation of lava viscosities before the discussion section. I detail my comments below, line by line, and hopefully the authors find them constructive to improve their paper.

Line 33: Here the author briefly discussed that lava viscosity can be visually roughly estimated by observing its surface velocity and behaviors, but the authors refer to Heslop et al. (1989) that is a study of super-elevated levees in a drained channel. I would add another reference, perhaps Lipman and Banks (1984) is more suitable because they visually estimated lava velocities a channels during the eruption.

Lipman, P. W. and Banks, N. G. (1987) A'a flow dynamics, 1984 Mauna Loa eruption, *US Geological Survey Professional Paper*, 1350, 1527-1567.

Line 34-35: The authors wrote: "... many of these were fueled by eruptions of alkaline mafic magma." and then refer to Baloga et al. (1995) but only one of the lava flow studied by Baloga et al. (1995) is alkalic (1801 Hualalai lava flow). I would add one or two more references here, perhaps from the works done on Nyiragongo lavas (Giordano et al., 2007; Morrison et al., 2020).

Line 62-68: This is the end of the introduction section, but the authors already present some of their results about lava viscosity. I would have kept this paragraph for the conclusion.

Line 79-83: Is there any indication of historical fast moving lava flows elsewhere on La Palma island? If yes, you could briefly describe them here.

Line 140: A "m" is missing at the end of "strea".

Line 173: Typo. Remove "The" before "these".

Line 174-179: Nepheline normative magmas are critically undersaturated in SiO₂ (high (Na₂O+K₂O)/SiO₂ ratios). They should also be olivine normative. I suggest to report the amount of normative nepheline in Table 1. I would rewrite the sentence as: "...which in turn shows that it belongs to the critically undersaturated magma series that differentiate with neutral..."

Line 187: Microphenocrysts or phenocrysts? What is the size range of the microphenocrysts? Can you provide an estimation of phenocrysts amount (in %). Could you add a few microscope photographs to Figure 5?

Line 189: What are these Fe-oxides? Magnetite? Hematite? Are they Ti-rich as well (titanomagnetite)?

Line 195: A parenthesis is missing to bracket Fig. 4.

Line 204-205: Refer to the supplementary info for the details of the clinopyroxene-liquid geothermometer and barometer. Why using cpx-melt thermometer to estimate the eruption temperature? Are there textural or geochemical evidences suggesting that cpx crystallize after olivine? Why olivine-melt equilibrium temperature is not calculated and used for eruption temperature?

Line 209: replace “cpx” by clinopyroxene.

Line 232: What is the temperature of crystallization onset? At what temperature form the first crystals? On Fig. 6, it seems that your viscosity measurements already deviate from the pure melt model at ~1200 °C.

Line 246-291: This is a great and detailed description of the Cumbre Vieja lava viscosity. The author did a good job at describing the melt viscosity from bulk rock compositions using the GRD model. Both dry and wet models appear good to me and the plagioclase-liquid hygrometer gives reasonable results for these mafic basalts. The calculated viscosity from the model is low, at least compared to a classic basalt, but given the composition of these basanites (low SiO₂ for high Na₂O+K₂O), the results are not too surprising.

The conclusion of these paragraphs is that the Cumbre Vieja magma as an ultra-low viscosity, perhaps in the range of komatiites or nepheline lavas. Like the author, I do believe the Cumbre Vieja melt viscosity was particularly low (given the chemical composition, the flow velocities and standing waves, etc.). However, the author did not mention the role of crystals in the bulk viscosity of the magma when using the GRD model.

The GRD model works very well to estimate the melt viscosity, i.e., pure liquid. But the apparent viscosity of a magma is defined as the product of the liquid viscosity by the relative effect of suspended crystals and vesicles. (Vesicles are present in various amounts in lava samples, and the effect of vesicles in the lava viscosity is dependent on the size and shape of the vesicles. The geometry of vesicles during lava emplacement is not known (although we should expect elongated shapes in such fast moving lava flows). Thus, their effect is often not considered.)

The crystal effect on the viscosity is directly related to the amount of suspended crystals in the flow, and it can more easily estimated. There are several methods to model the relative effect of crystals in the melt, there is an extensive literature about this. A model that is often used is the Krieger and Dougherty's equation (see Mueller et al., 2010). This model is dependent of the maximum crystal packing and coefficients dependent of the crystal shapes. Usually, we can simplify olivine and oxide crystals as spheres, whereas plagioclase crystals can be considered as needles. I suggest the author to weight the GRD model by the amount of phenocryst in their sample (olivine and clinopyroxene).

Comments on the Discussion section:

Line 326-327: Is there any field evidence to suggest that the channel was rectangular in cross-section? Why not considering a U-shape or semi-circular shape?

Line 331: You only have 10 cm uncertainties of the channel depth? That seems very small, perhaps it is a typo?

Line 334: The flow depth was only 0.3 meter? This also seems pretty low... How did you estimate the flow depth? This is not explained in the supplementary information

Line 334: 2700 kg/m³ is the density of vesicle-free basaltic magmas. However, the lava was most likely containing vesicles. What is the resulting Re number assuming 10%, 20%, 30% etc, vesicles? The author

discussed the effect of channel geometry and lava viscosity of the calculated Re , but did not discussed the effect of density.

Line 385: Gives the values of velocity from the videos? Would be interesting to compare these with velocity where standing waves have been observed.

Line 395: The value of Fr and high Re are not unprecedented. Observed standing waves indicated Froude number perhaps as high as 3 (Le Moigne et al., 2020). I am sorry for introducing my work here, but I do not know other good examples.

Line 399: Again I would revise this statement. Yes, the melt viscosity is very low, but the magma viscosity was certainly higher because of the crystal content at eruption temperature.

Line 416: In the references, add the recent work of Morrison et al. (2020) on the Nyiragongo and Nyamuragira lava viscosities. They estimated lava viscosities on the order of tens of Pa.s.

Morrison, A., A. Whittington, B. Smets, M. Kervyn, and A. Sehlke (2020), The rheology of crystallizing basaltic lavas from Nyiragongo and Nyamuragira volcanoes, D.R.C., *Volcanica*, 3 (1), 1–28, doi:10.30909/vol.03.01.0128.

Figure caption 5: Typo in line 4. A “i” is missing for “microlite”.

Submitted by Yannick Le Moigne
Simon Fraser University
Vancouver, BC

Comments of Reviewer 1 (Yannick Le Moigne)

This is a study of the viscosity properties of the recent basanite eruption of Cumbre Vieja volcano, La Palma, both from field observations (standing waves) and lab experiments. Overall, I thoroughly enjoyed reading this paper and I consider to be appropriate to publish it in Nature Communication. Eruption of mafic alkaline lavas are rare, and the Cumbre Vieja eruption provide an excellent opportunity to gain insight on the rheological properties of these lavas. In my opinion, the manuscript only need moderate revisions. My main comments concern the calculation of lava viscosities before the discussion section. I detail my comments below, line by line, and hopefully the authors find them constructive to improve their paper.

We thank Dr. Le Moigne for his input on our manuscript. We have addressed all of the comments to the best of our ability, and think the manuscript is consequently greatly improved. The calculation of lava viscosities now accounts for the possible effects of included crystals, an issue that was also noted by reviewer 2. Our new data and assessments of the effects of crystals on rheology show that adjusted viscosities will not shift much, at most a factor of two higher, and this relative effect applies only in the case that the magma had crystallized to levels that are not evidenced in our experiments nor supported by the geothermobarometry data. We detail this point in the passages below.

Line 33: Here the author briefly discussed that lava viscosity can be visually roughly estimated by observing its surface velocity and behaviors, but the authors refer to Heslop et al. (1989) that is a study of super- elevated levees in a drained channel. I would add another reference, perhaps Lipman and Banks (1984) is more suitable because they visually estimated lava velocities a channels during the eruption.

Lipman, P. W. and Banks, N. G. (1987) A'a flow dynamics, 1984 Mauna Loa eruption, US Geological Survey Professional Paper, 1350, 1527-1567.

We appreciate this suggestion and have added the above-cited reference. We note also that the wording of the text has been slightly modified to better accommodate the Heslop et al. (1989) reference and the point that "lava" may be both actively flowing material and also the solidified remains of a flow. The point we emphasize that flow rheology can be qualitatively visually estimated applies to both of these states of lava and hence the Heslop et al. (1989) reference should remain. The revised text is on lines 30-33 of the new manuscript and reads as follows (new wording in bold):

*"...viscosity can be summarily assessed during eruptions with the "naked eye", by for example observing how "fluid" the lava **appears** and estimating flow velocities as it surfaces⁸⁻¹⁰."*

Line 34-35: The authors wrote: “... many of these were fueled by eruptions of alkaline mafic magma.” and then refer to Baloga et al. (1995) but only one of the lava flow studied by Baloga et al. (1995) is alkalic (1801 Hualalai lava flow). I would add one or two more references here, perhaps from the works done on Nyiragongo lavas (Giordano et al., 2007; Morrison et al., 2020).

We appreciate this suggestion and have added the two suggested references; we have of course retained the Baloga et al. (1995) reference in this section.

Line 62-68: This is the end of the introduction section, but the authors already present some of their results about lava viscosity. I would have kept this paragraph for the conclusion.

We very much appreciate this feedback. We agree that the quantitative information in this passage should be withheld for presentation in later sections. Thus, we have removed the numerical values of viscosity from this section. However, we believe that it is important to maintain the general writing style and context of this section because this leads the reader directly and intuitively along to the next part of the paper. We have used this method of writing salient results in the introduction before in *Nature Communications* articles (see e.g., Castro et al., 2016). In these cases, and here, the goal is to maintain interest in the work and to keep the reader engaged. Thus, we have not removed nor modified the last section of the paragraph, which makes the important point that Cumbre Vieja produced magma whose physical properties were unlike those of typical basalts. The slightly revised text appears on lines 62-67 of the revised manuscript:

*“...our results show that the viscosity of thoroughly degassed Cumbre Vieja lava could have **been exceedingly low (tens of Pa sec)** across a range of permissible eruption temperatures. These physical properties position the Cumbre Vieja basanite in a rarely observed behavioral class, in which inertial effects on the flow were important, suggesting that some lava flow facies were probably turbulent and subject to hydraulic jumps^{20,21}.”*

Line 79-83: Is there any indication of historical fast moving lava flows elsewhere on La Palma island? If yes, you could briefly describe them here.

Thank you for this question. We have scoured the literature and found references to “fluid lavas” during La Palma’s historical 1677 San Antonio and 1971 Teneguía eruptions; this information is given in a paper by Carracedo et al (1996), which we now cite in the revised manuscript. Unfortunately we did not find flow velocities reported for these historical events. Nonetheless, we think that the inclusion of the Carracedo et al (1996) paper will lead interested readers in the right direction and, more importantly, indicate that historical basanite eruptions on La Palma behaved much like the 2021 event. We have slightly adjusted the text on lines 79-84 to indicate both the inclusion of the Carracedo et al. reference and the brief indications that other lavas from La Palma (Cumbre Vieja) were described as “fluid” with the

implication being that these too were of low viscosity:

*"Basanite lava flows emplaced during these historical eruptions (e.g., the 1677 San Antonio and 1971 Teneguía events) are notable for their **observed "fluid" behaviour**²⁴ and resemble those of the 2021 event in that they are long (3-8 km) and slender, **characteristics of which** manifest both magma transport properties (e.g., viscosity) and direct emplacement paths to the sea (Fig. 1)."*

Line 140: A "m" is missing at the end of "strea".

Thank you for this correction; we have made the change.

Line 173: Typo. Remove "The" before "these".

Thank you for this correction; we have made the change.

Line 174-179: Nepheline normative magmas are critically undersaturated in SiO₂ (high (Na₂O+K₂O)/SiO₂ ratios). They should also be olivine normative. I suggest to report the amount of normative nepheline in Table 1. I would rewrite the sentence as: *"...which in turn shows that it belongs to the critically undersaturated magma series that differentiate with neutral..."*

We appreciate this comment. We have changed the text to reflect that the magma is critically undersaturated in SiO₂. We also now refer to Table 1 in the text, where the reader can find the normative amount of Nepheline. Note finally that the mineralogical description of the tephra starting material along with our reference to the recently published petrological work of Pankhurst et al (2022), provide unequivocal evidence of olivine in the mode of this magma, which precludes the need to demonstrate a normative calculation for this mineral. The lightly modified text now appears on lines 180-182 of the revised manuscript:

*"...which in turn places it in the family of **critically SiO₂-undersaturated** alkaline magmas that differentiate with neutral, or only slight SiO₂ enrichments as the system crystallizes¹³."*

Line 187: Microphenocrysts or phenocrysts? What is the size range of the microphenocrysts? Can you provide an estimation of phenocrysts amount (in %). Could you add a few microscope photographs to Figure 5?

We thank the reviewer for these questions. In our survey of tephra samples, we did not come across crystals that we would classify as true phenocrysts, ie., crystals easily observable with the un-aided eye (>1 mm) and thus we describe the somewhat larger crystals (in comparison to groundmass microlites) as microphenocrysts. We now indicate a size range (~40-500 µm) of these crystals in the referenced sentence. The abundance of these crystals was already given in the original text (~5-10 vol.%).

We appreciate the suggestion to add microscope images to Figure 5. We feel that the Figure 5 is simple and should be left unmodified. based on the fact that none of our analysis relied on traditional light microscopy and in light of the fact that we cite Pankhurst et al. (2022) who survey the microscopic characteristics of tephra including providing microscope images, we have refrained from fulfilling this request.

Line 189: What are these Fe-oxides? Magnetite? Hematite? Are they Ti-rich as well (titanomagnetite)?

We appreciate this question. On the basis of new microprobe analyses, we have determined that the Fe-oxides are uniformly titanomagnetite. We did not identify any other oxide phase. We have replaced the word “Fe-oxides” with “titanomagnetite” in the above-cited text line (189), and have added a small sentence to the Fig. 5 caption to indicate that oxides and silicates pictured in the BSE images were identified by EPMA analyses; the sentence reads:

“All silicate and oxide minerals were identified by EPMA analysis”

Line 195: A parenthesis is missing to bracket Fig. 4.

Thank you. This has been corrected.

Line 204-205: Refer to the supplementary info for the details of the clinopyroxene-liquid geothermometer and barometer. Why using cpx-melt thermometer to estimate the eruption temperature? Are there textural or geochemical evidences suggesting that cpx crystallize after olivine? Why olivine-melt equilibrium temperature is not calculated and used for eruption temperature?

We thank the reviewer for these insightful questions and the suggestion to refer to the supplementary information. We now refer to the supplementary information in the revised section. As to the question of why we used the clinopyroxene-liquid geothermometer, we draw the reviewer’s attention to the first sentence of this paragraph, which indicates that the abundance of clinopyroxene in contact with fresh glass constitutes a good basis for application of the geothermometer. We do not have any better rationale to add, and given the fact that cpx-glass is a widely used tool—including applications of this thermometer by other authors on historical Cumbre Vieja basanites (see ref. 23 Weis et al 2015)—our use of it should not seem out of the ordinary or unjustified.

We do not see textural evidence for a crystallization sequence as regards clinopyroxene and olivine in the basanite tephra we studied. We therefore cannot assess a crystallization order amongst olivine, clinopyroxene and oxide minerals. This question, while interesting, is largely beyond the scope of our paper, even though we do recognize the importance as regards geothermometry data.

We appreciate the suggestion to look at results from the olivine-liquid geochemical data. Accordingly, we have applied various published olivine – liquid thermometers to our dataset including models using solely the melt (glass) composition for the calculations (Heltz & Thornber 1987, Yang et al. 1996, Fabbriozio & Spillar 2020; Supplementary Information), and those using both, liquid and olivine compositions (ref. 36, Beattie 1993; ref. 32, Putirka 2008 eq. 22). The results indicate a broad range of temperatures: from around ~1100 °C up to ~1185 °C, dependent on the sample, and conditions such as pressure (7 – 10kbar), and H₂O content (0 – 0.8 wt.%). Results are also highly dependent on which model was applied. We consider the models of Beattie (1993) and equation 22 of Putirka (2008) to be best suited for anhydrous and hydrous melts, respectively, as these are widely used in the published literature, and include a large number of olivine – liquid compositions in the calibrations. These models yield anhydrous temperatures of 1160 – 1184 °C, and 1100 – 1144 °C for hydrous conditions, being roughly in line with temperatures estimated by cpx – liq thermometry (ie., those included in the original draft).

However, equilibrium tests based on the comparison of predicted vs. measured $K_D(\text{Fe-Mg})^{\text{ol-liq}}$ partition coefficients (e.g. Putirka 2008) indicate that the olivine crystals of the studied samples are not equilibrated with the coexisting melt. Disequilibrium is further evidenced by slight edge zoning and weak resorption of olivine crystals in the studied samples (cf. Fig. 5c). Therefore, we decided not to use the calculated ol-glass temperature estimates in assessments of permissible melt viscosity based on rheometry experiments. However, these new data (and their shortcomings related to disequilibrium) are now noted in the revised manuscript. For example, we have included short paragraphs in the main text and in the Methods section describing the application of olivine – liquid thermometry to our data. In the main text of the revised manuscript, appearing on lines 223-231, one finds the following new text:

"Application of olivine-liquid geothermometers^{32,36} (eq. 22 in ref. 32) to the tephra samples yields a broad temperature range (~1100-1185 °C) depending on pressure (7 - 10 kbar) and H₂O content (0 - 0.8 wt.%; Methods). However, owing to large (>30%) discrepancies between measured and predicted olivine-melt partition coefficients, $K_D(\text{Fe-Mg})^{\text{ol-liq}}$, equilibrium conditions between olivine and coexisting melt were not likely met. Therefore, we consider the temperatures (~1150-1200°C) calculated by clinopyroxene-liquid thermometry to be more robust and use these values in subsequent assessments of permissible viscosities deriving from rheometry experiments."

In the methods we have added the following text (see lines 649-666 of the revised draft):

" In addition to clinopyroxene–liquid geothermobarometry, olivine (ol)–liquid (liq) pairs were investigated to determine magma temperature by applying two different published ol–liq geothermometers^{28,30}. For this purpose, individual olivine rim and

core compositions were used in concert with averaged glass compositions of the different samples. For anhydrous conditions, the model of ref. 36 was applied while for hydrous conditions (0.8 wt.% H₂O) ref. 32 (eq. 22 therein) was used. These models yield temperatures of 1160 – 1184 °C (anhydrous conditions), and 1100 – 1144 °C (hydrous conditions), which are roughly in line with temperatures estimated by cpx–liq thermometry. However, as indicated by a significant deviation (>30%) of the majority of the measured $K_D(\text{Fe-Mg})^{\text{ol}}$ values obtained from olivine compositions compared to the predicted $K_D(\text{Fe-Mg})^{\text{liq}}$ values calculated from the glass composition²⁸, it cannot be verified that the olivine crystals were in equilibrium with the coexisting melt. Disequilibrium conditions are further supported by apparent zoning as indicated by bright and resorbed rims of olivine crystals in backscattered electron images (Fig. 5c). Therefore, temperatures calculated from olivine–liquid thermometers, even though broadly in line with those obtained by clinopyroxene–liquid compositions, were not considered for the assessment of probable eruption temperatures.”

Line 209: replace “cpx” by clinopyroxene.

Thank you for this suggestion. We have corrected this in the revised manuscript.

Line 232: What is the temperature of crystallization onset? At what temperature form the first crystals? On Fig. 6, it seems that your viscosity measurements already deviate from the pure melt model at ~1200 °C.

We thank the reviewer for these insightful questions. As we mentioned in an earlier response, we didn’t find textural evidence for a crystallization sequence, none that could be used to infer a potential crystallization onset. However, our “1-atm heating experiments”, shown in Fig. 6 as a series of BSE images, indicate a potential crystallization start at about 1265°C. In this experiment the volume percent of crystals is very low (<0.1 wt.%), which could be interpreted as an “onset” of crystallization. We are not sure how or if we should integrate this point into the discussion, as this temperature and low crystallinity are outside (ie., higher than) the implied eruption/magma temperatures (~1150-1200°C) and thus questions about crystallinity at such high T’s may distract from the salient findings of our study. We have however modified the text slightly to draw the attention to the point that crystals were likely present in the relevant range of experimental *and* eruptive conditions, and that the crystallinities measured in the 1-atm experiments are likely “*minimum*” values due to the short heating durations applied. The modified text appears on lines 251-256 of the revised manuscript:

“These 1-atm experiments furthermore indicate relatively subtle (<0.1 to ~4 vol.%) crystallinities in melts at higher T’s (~1163-1265°C). However, we consider these crystal contents to reflect minimum values given that the short durations of these experiments (10’s of mins) may not have fostered attainment of textural equilibrium at a given T.”

As pertains to the reviewer’s 2nd point, namely that it seems the viscosity

measurements deviate from the pure-melt model curves already at 1200°C, we must again thank the reviewer for recognizing this. We agree with the reviewer that this discrepancy could reflect the effect of crystals on bulk viscosity at 1200°C and extending to lower T's. Because 1-atm crystallization experiments (r.h.s. Fig. 6) were of short duration (15-30 mins), it is hard to say that these represent equilibrium crystallinities. We think it is more likely that they represent minimum values that could have likely increased in the long (hours to days) rheology experiments. Taking the advice of this reviewer, as well as reviewer no. 2, we have calculated the effect of crystals on the rheology using textural measurements made on the tephra samples and applying the model of Klein et al. (2018). Our approach and the results of these textural measurements are described in the revised manuscript results and methods sections. We draw your attention specifically to the revised results text, which has now been newly written and adjusted to reflect our treatment of crystals in the experimental melts and appears on lines 282-306 of the revised manuscript:

“...which could reflect the presence of crystals in the experimental melt, and the tendency of crystals to increase the bulk stress and therefore the effective viscosity of the magmatic suspension³⁷⁻³⁹. We have investigated the feasibility of this interpretation by calculating the relative viscosity (μ_{rel})^{31,37}—the viscosity of the crystal-melt suspension normalized by the pure-melt viscosity ($\mu_{rel}=\mu_{sus}/\mu_{melt}$)—of a dry basanite magma containing modest (~6 to 16 vol. %) amounts of crystals (Supp. Info.). These crystal contents are consistent with what is observed in the tephra samples (Fig. 5; Supp. Info.), and are used as a guide to investigate potential effects of crystallinity on rheology. The crystal contents, along with crystal size and shape distributions needed to perform the rheological calculations (via ref. 31), derive from textural measurements that we made on six tephra clasts sub-sampled from the bulk ash that also served as experimental starting material (Fig. 5; Supp. Info.). These calculations show that the effects of the tephra's crystal cargo—taken as a proxy for what could have conceivably been present in the freshly erupted lavas—on the viscosity of the suspension is small, as reflected by relative viscosities of only ~ 1.5 to 1.9 (Table 1). These relative viscosities mean that all crystal-free estimates of melt viscosity, including the estimates for anhydrous basanite derived from the GRD model (Fig. 6), could profitably be “adjusted up” ie., multiplied by the relative viscosity to derive possible effective crystal-bearing melt viscosities. Thus, applying these corrections to the GRD model viscosities in the experimental range of 1200 to 1150 °C (Fig. 6) will close the gap between experimental viscosities and the adjusted GRD values. In other words the offset between measured and predicted anhydrous melt viscosities are likely due to the presence of a small quantity of crystals in the experimental melts.”

Line 246-291: This is a great and detailed description of the Cumbre Vieja lava viscosity. The author did a good job at describing the melt viscosity from bulk rock compositions using the GRD model. Both dry and wet models appear good to me and the plagioclase-liquid hygrometer gives reasonable results for these mafic basalts. The calculated viscosity from the model is low, at least

compared to a classic basalt, but given the composition of these basanites (low SiO₂ for high Na₂O+K₂O), the results are not too surprising.

We appreciate these comments.

The conclusion of these paragraphs is that the Cumbre Vieja magma as an ultra-low viscosity, perhaps in the range of komatiites or nepheline lavas. Like the author, I do believe the Cumbre Vieja melt viscosity was particularly low (given the chemical composition, the flow velocities and standing waves, etc.). However, the author did not mention the role of crystals in the bulk viscosity of the magma when using the GRD model.

Thank you for these great insights and the point that we should address the effect of crystals on apparent viscosity of the Cumbre Vieja magma. The revised draft now contains the requested information, particularly in the revised results section at the position referred to by the reviewer (presentation of “Viscosity measurements”). As we described in our response to the reviewer’s previous question regarding line 232 above, we have performed textural measurements on tephra clasts to quantify crystallinities, size distributions, and crystal shapes, in order have the inputs to the rheological model developed by Klein et al. (2018). This model is an elaboration on the Mueller et al. (2010) suspension rheology model and holds the advantage that it calculates all important physical parameters (e.g., maximum packing fraction and polydispersity) from textural measurements. As detailed above and in the revised manuscript, results of these calculations show that the effects of modest crystal contents—values that are evidenced by, and derived specifically from tephra clast textures—will only shift the pure-melt viscosities to slightly higher values. More specifically, the relative viscosities (ie., the suspension viscosity normalized by the pure-melt viscosity) are only 1.3 to 1.9. These represent shift factors that amount to subtle increases in effective viscosity over the pure-melt values. Thus, our conclusions remain the same and “ultra-low” viscosity is a justified and appropriate description of Cumbre Vieja’s magma. The following revised text highlights the additional changes made (bold font) in order to clarify differences between crystal-free and crystal-bearing values, and to better integrate the results of relative viscosity calculations; this new text appears on lines 314-340 of the revised manuscript:

*“Of the viscosity data, the most relevant to understanding the rheological state of near-vent, freshly emergent basanite magma, are those falling within the estimated eruption temperature range (~1150-1200°C; Table 1). Thus, effective **anhydrous** magma viscosities are about 50 Pa s to a little over 160 Pa s. These values, while significantly lower than recent tholeiitic lava eruptions (~250-1150 Pa s)^{2,42}, are quite comparable with field-based μ estimates of historical Hawaiian lavas that exhibited exceptionally fluid, near-vent behavior, including supercritical flow phenomena^{8,43} (~10² Pa s). In the high likelihood that the Cumbre Vieja basanite was even slightly hydrous—conditions that **are** supported by the explosive nature of the eruption, plagioclase-glass hygrometry³⁵ (Table 1; Supp. Info.), **the bulk basanite tephra’s***

*loss-on-ignition (L.O.I.; Table 1), and previous findings on other Cumbre Vieja basanites suggesting magmatic H₂O contents of ~ 0.5-1.0 wt.%^{23,34,44} —then both **pure- and crystal-bearing melt** viscosities would be substantially lower. Calculations using the GRD model (Fig. 6), indeed indicate that hydrous basanite melt (**crystal-free**) containing ~1.0 wt.% will be more than an order of magnitude less viscous than the anhydrous melt from 1150°C to 1200°C (e.g., **as shown by the hydrous melt μ GRD curves in Fig. 6; ~ 15 Pa s to ~ 7 Pa s**). With ~0.5 wt.% H₂O the viscosity is only slightly higher than the 1.0 wt.% melt, varying from about 23 to ~11 Pa s from 1150°C to 1200°C. These GRD-determined values bracket what the permissible crystal-free melt viscosity of the 2021 Cumbre Vieja magma would be: basanite melt with an estimated 0.8 wt.% H₂O is just ~18 to 9 Pa s across the eruptive temperature range. **In the likely case that a small complement of crystals were present in the hydrous melt (e.g., Fig. 5, 6), the effective magma viscosities^{31,37} would be slightly higher (1.3-1.9) than the pure-melt values, but ultimately would not exceed a few tens of Pa s.** Such ultralow viscosities are not typical of basaltic magma^{4,9}, and instead, resemble those of the ultramafic komatiite melts³⁹ and nephelinite lavas¹⁷.”*

The GRD model works very well to estimate the melt viscosity, i.e., pure liquid. But the apparent viscosity of a magma is defined as the product of the liquid viscosity by the relative effect of suspended crystals and vesicles. (Vesicles are present in various amounts in lava samples, and the effect of vesicles in the lava viscosity is dependent on the size and shape of the vesicles. The geometry of vesicles during lava emplacement is not known (although we should expect elongated shapes in such fast moving lava flows). Thus, their effect is often not considered.))

We thank the reviewer for these comments. We have addressed the issue of suspended crystals in the previous two responses and agree that the geometry and abundance of bubbles in the lava is unknown. We therefore have not made revisions to this section in this regard.

The crystal effect on the viscosity is directly related to the amount of suspended crystals in the flow, and it can more easily estimated. There are several methods to model the relative effect of crystals in the melt, there is an extensive literature about this. A model that is often used is the Krieger and Dougherty's equation (see Mueller et al., 2010). This model is dependent of the maximum crystal packing and coefficients dependent of the crystal shapes. Usually, we can simplify olivine and oxide crystals as spheres, whereas plagioclase crystals can be considered as needles. I suggest the author to weight the GRD model by the amount of phenocryst in their sample (olivine and clinopyroxene).

Thank you. We have applied the model of Klein et al. (2018), which indeed accounts for variable crystal size and shape. With these new data, we feel our conclusions are fully justified and more robust. We appreciate the careful suggestions of this and

the other reviewer on this issue.

Comments on the Discussion section:^[11]_[SEP]

Line 326-327: Is there any field evidence to suggest that the channel was rectangular in cross-section? Why not considering a U-shape or semi-circular shape?

We appreciate these questions. Because remote video was our only source of information, it was very difficult to determine both the scale and geometry of these lava cascades. In short, we were not able to confirm evidence for a semi-circular shape channel. Furthermore, because the flows are so proximal and their effusion is the product of overflowing and being fed by fountains, they did not enter pre-existing structures that were significantly deep, nor did they grow levees or crust over as they flowed down slope. We therefore cannot confirm that a U-shape or semi-circular channel geometry is a better choice. What is true about the flows is that they are thin and wide, akin to sheet flows. As the point of this section is to determine order-of-magnitude estimates of the Re , and to do this accounting for evidence that the lavas are clearly relatively thin, sheet like flows, we have chosen a shallow rectangular cross sectional channel to calculate the hydraulic diameter. We have adjusted the text to better express our choice of a thin rectangular channel; this can be read on lines 374-381 of the revised manuscript:

"... D is the hydraulic diameter, or characteristic length defined for open rectangular channels as four times the product of the channel depth and width divided by the wetted perimeter ($P=2h+w$)^{8,21,46}. As we have few constraints on the cross-sectional geometry of the cascade channels (Fig. 2), and owing to the fact that these proximal flows did not appear to enter pre-existing mature lava structures, we will operate on the assumption that these are relatively thin, tabular flows akin to sheets moving down channels of shallow rectangular cross section⁸."

Line 331: You only have 10 cm uncertainties of the channel depth? That seems very small, perhaps it is a typo?

Thank you for this request for clarification. We have indeed reported what we think is the uncertainty in channel depth based on these lavas having an average thickness of about 30 cm. We agree that this seems low, at least the uncertainty is perhaps underestimated and unjustified given what we can estimate from video footage. Thus, we have adjusted this up to ± 20 cm in the revised text. This thickness uncertainty accounts for flows that may start out very thin and inflate a little, or may represent the uncertainty stemming from crust formation and subsequent breakup on lavas filmed in the field and posted as online videos. As described below in the following response, we used some of these online videos to determine possible *minimum* flow thicknesses of the order 30 cm. The Supplementary Information has been furthermore updated to reflect exactly how we determined flow thicknesses using the thin lavas erupted and filmed by an unknown source on 25 November

2021.

Line 334: The flow depth was only 0.3 meter? This also seems pretty low... How did you estimate the flow depth? This is not explained in the supplementary information

Thank you for these questions. As this also relates to the previous question on the uncertainties of flow depths, we will reiterate that these estimates derived from comparing the thickness of the “thinnest” observed flows on La Palma in videos (25 November 2021) to substrate clastic material that we assumed, based on textural appearance, was in the ash to lapilli size range (~mm’s). Numerous other online videos show very thin flows erupted in the 2021 event, particularly on the 25th of November 2021. We estimated the thickness of these flows by comparing the flow’s vertical dimension to ash and lapilli sized particles (mm’s) over which it flowed, in addition to shrubs that the flow passed by (the same kind of shrub was seen in El Paso where we made our field observations and these were on the order of 0.5 m tall). We recognize that this is only an approximate scaling, but it is all we had to work with due to our non-existent access to the active lavas. We have now clarified in the revised Supplementary Information the means by which we estimated both flow thickness (or channel depths) and widths, in addition to adding other information on how the velocities of flows observed to produce standing waves were determined. Unfortunately the copyright owner of these videos referenced for lava flow scaling is unknown and therefore we cannot include these videos in the supplementary information.

Lastly, we have looked at archival videos of lavas from the 1971 basanite eruption of Teneguía as well, and can confirm that these basanite flows, which have nearly an identical composition and presumed eruption temperature to the 2021 flows, indeed emerged with very thin (~30 cm) profiles.

Line 334: 2700 kg/m³ is the density of vesicle-free basaltic magmas. However, the lava was most likely containing vesicles. What is the resulting Re number assuming 10%, 20%, 30% etc, vesicles? The author discussed the effect of channel geometry and lava viscosity of the calculated Re, but did not discussed the effect of density.

This is a very good point and we appreciate the opportunity to address this. We have now added a new paragraph discussing the effect of reduced magma density as would occur for bubbly magma. We only address the case of 30 vol.% bubbles but do give a general description of the effect on *Re*. In addition, we speculate on what the likely effect of thicker flows would be in comparison to the bubble effect, as this will more than counteract the effect of bubbles (e.g., considering a marginally thicker lava of 0.5 m). Finally, we now speculate on the effect of bubbles on magma rheology.

Notwithstanding our response to this inquiry, it is important to note that we have

no constraints on the bubble content of the lava in the cascades. Furthermore, a thorough investigation of the entire parameter space underpinning possible Re number regimes is very much beyond the scope of this work. Our goal was to establish what the viscosity of the magma was in its freshly emergent state, and we have succeeded in that. An ancillary goal was to explore *potential* implications of ultralow viscosity magma, particularly those that would lead to the odd hydrodynamic behavior that the lava exhibited in the field. This too, was accomplished in the original and now revised manuscript.

Please see the following revised text as it appears on lines 400-409 of the revised manuscript:

“The effect of reduced bulk magma density, as caused by the presence of bubbles, would lower Re estimates by an amount equal to the fractional bubble concentration. The reduced density effect of bubbly magma would however be readily offset by the flows having greater thickness. For example a flow of 30% porosity but of a marginally higher 0.5 m thickness (in contrast to 0.3 m considered earlier) will have a Re in excess of 2200. We have furthermore not considered the potential effects of bubbles on the effective viscosity due to a lack of constraints on lava vesicularity during eruption of the cascade lava. At the high implied flow and shear rates, bubbles would be highly deformed, leading to a high capillary number regime and consequently negligible effects on the effective viscosity^{2,6}.”

Line 385: Gives the values of velocity from the videos? Would be interesting to compare these with velocity where standing waves have been observed.

We thank the reviewer for this request. We have expanded upon our methodology in the Supplementary Information to describe how we determined the flow velocities (~ 7 m/sec) of the standing wave lavas pictured in Fig. 3. We have added this result to the text in this section. Our Froude number calculations in the revised manuscript now account for this velocity (ie., yielding $Fr=2.2$), in addition to stating the Fr 's that apply for a reasonable range of flow velocities. Particularly, we include bracketing upper velocity (~ 10 m/sec; $Fr=3.2$) and lower values (~ 5 m/sec; $Fr=1.6$) in order to cover probable measurement error. Importantly, Froude numbers at the lower velocities still indicate super-critical flow. The revised text section, appearing on lines 448-456, more succinctly explains Fr calculations. The text is as follows in the revised manuscript:

“Here v is the velocity, g is the gravity constant, and h is the flow depth. Video analysis of the flow depicted in Fig. 3 (Supp. Info.), suggests an approximate velocity of 7 m sec⁻¹. If this flow were 1 m thick, Fr would be about 2.2. Clearly great uncertainties exist in this flow's dimensions and therefore velocity estimates. Nonetheless uniformly high Fr numbers (1.6 - 3.2) are implied for a range of reasonable velocities ($\sim 5 - 10$ m sec⁻¹). Generally for $Fr > 1$, the flow will occur in a supercritical regime characterized by hydraulic jumps^{17,18} and the persistence of gravity waves whose relative speeds are much less than the overall flow velocity.”

Line 395: The value of Fr and high Re are not unprecedented. Observed standing waves indicated Froude number perhaps as high as 3 (Le Moigne et al., 2020). I am sorry for introducing my work here, but I do not know other good examples.

We thank the reviewer for this comment. This is absolutely correct. We have adjusted the text to indicate that the values are “uncommon” and not unprecedented. Le Moigne et al., 2020 has now been inserted into the statement in the revised manuscript.

Line 399: Again I would revise this statement. Yes, the melt viscosity is very low, but the magma viscosity was certainly higher because of the crystal content at eruption temperature.

We thank the reviewer for this suggestion. We will however leave the statement unchanged due to new data (textural assessment and rheology calculations) that indicate only *subtle* upward shifts (1.3-1.9 times) in effective viscosity. In addition we think that it is possible that pure-melt viscosities could also be relevant in the event that the magma erupted at or above the upper end of the estimated temperature range (1200°C), or if the magma were more hydrous than we estimated, or were essentially crystal poor as it rose through the crust to the point of eruption.

Line 416: In the references, add the recent work of Morrison et al. (2020) on the Nyiragongo and Nyamuragira lava viscosities. They estimated lava viscosities on the order of tens of Pa.s. Morrison, A., A. Whittington, B. Smets, M. Kervyn, and A. Sehlke (2020), The rheology of crystallizing basaltic lavas from Nyiragongo and Nyamuragira volcanoes, D.R.C., *Volcanica*, 3 (1), 1-28, doi:10.30909/vol.03.01.0128.

We thank the reviewer for this comment. We have added this reference to bolster the point made early in the text about alkaline magmas (also in response to this reviewer’s earlier comment about the Baloga et al. 1995 reference on lines 34-35).

Figure caption 5: Typo in line 4. A “i” is missing for “microlite”.

We thank the reviewer for this comment. We have corrected the figure 5 caption as indicated above.

Comments of Reviewer 2 (Rebecca deGraffenried)

The manuscript titled “Eruption of ultralow-viscosity basanite magma at Cumbre Vieja, La Palma, Canary Islands” by Castro and Feisel presents measurements of the viscosity of material erupted from Cumbre Vieja in 2021, estimates of eruption temperature and melt water content from cpx-liquid thermobarometry and plag-liquid hygrometry, and observations of channelized lava flow dynamics. The authors demonstrate that the fast-moving lava flows that devastated the nearby community could be the result of ultralow viscosity lava and that this ultralow viscosity was reflected in the observed supercritical flow features. This type of work is critical for hazard assessment and mitigation, as fast-moving lava flows are a serious threat to nearby populations, thus this work is significant for the field of volcanology.

We thank the reviewer for their attention to detail in their comments and for this nice summary of our work.

I find this work to be well-written and compelling, with just a few areas for wording clarification and typo correction. The data appear to be sound and collected in an appropriate manner.

We thank the reviewer for this comment. We’ve endeavored to answer all of their following comments with rigor and care.

However, with regards to the interpretation of the results, I do have three general points that I think would benefit from additional consideration from the authors:

1) How representative is the ash of the material that fed the fast-moving lava flows, particularly in terms of crystal microtexture? From the description of the eruption that the authors provide, the ash is primarily sourced from vents that had a “duller, orange- incandescent” lava effusion, in contrast with the vents of interest that effused lava that was “extremely bright” and “emitted a white radiance” (lines 114-115). This question ties into my next two points.

This is a great question. It remains unclear what the microtexture of the freshly emergent lava is, since no samples were made available to us when we were in the field (due to the exclusion zone), nor are we aware of any rapidly quenched samples collected from the lava cascades that could provide confirmation of the lava texture. The point of our not having access to lavas was made in the original draft (lines 146-147) and it is for this reason that we used tephra as a starting material for the experiments. We expect no compositional differences between the tephra we used and the lavas, as Pankhurst et al (2022)—a paper we cite in the revised draft on lines 172-173—also measured bulk rock compositions of Cumbre Vieja’s eruption products and found identical compositions (within analytical uncertainty) to our tephra samples. To clarify this point, we now add a reference (to Pankhurst et al,

2022) and short mention of the compositional similarity on lines 174-176 of the revised draft. All of this indicates that our use of tephra to study the *emergent* lava rheology is compositionally self-consistent, but, it remains uncertain what the crystallinity of the newly emergent lavas in the cascades was. The new sentence appears on lines 177-179 of the revised draft as follows:

“These data indicate that the magma is a basanite (Fig. 4), corroborating the results of previously published geochemical and petrological analyses on Cumbre Vieja’s 2021 lava¹⁸. “

2) If the ash is indeed representative of all material being erupted, I am not certain that the rheometry measurements capture the bulk properties of the lava. From the BSE images in Figure 5, the crystal cargo was primarily dominated by elongate plag and pyroxene crystals, up to 30 vol. %. Since the size of the ash is quite small, it can safely be assumed that the texture of the magma as it erupted was quenched, with little modification from cooling. In contrast, the BSE images of the melts quenched from the rheometer experiments in Figure 6 show primarily equant oxides. The most comparable to expected eruption temperature experiments shown are the 1213 and 11630C experiments, but those show vastly different textures to the erupted products, in terms of overall crystal content and crystal shape. Although the crystal content in the natural products is relatively low, the dominance of high aspect ratio crystals could induce changes to viscosity at much lower volume fraction (e.g., Mueller et al., 2011, Geophysical Research Letters; Cimarelli et al., 2011, Geochemistry, Geophysics, Geosystems). I would like to see whether the increase in measured viscosity relative to the crystal-free value predicted by GRD is comparable between the experiments and the theoretical values predicted by models such as Costa et al. 2009 (Geochemistry, Geophysics, Geosystems) for elongate particles.

We thank the reviewer for these comments and questions. These are very important points and we appreciate that both reviewers have in particular raised questions regarding the effect of crystals on the rheology of the Cumbre Vieja magma. As we noted in our response above, we do not expect any complications with using tephra as a rheological experimental starting material in terms of its *composition*. The compositions between lava and tephra do not differ. However, it remains to be established whether the freshly emergent lava chronicled in this paper and in the videos of the lava cascades contained the same or a similar complement of crystals as the tephtras. As the reviewer notes, and as we originally stated in the manuscript, the lava was noticeably more radiant than tephra fountains, which implies that this magma may have been hotter, and by inference less crystalline than the tephtras. That said, it is beyond our abilities to prove this given our limited access to samples and this consequently means that we cannot confirm that **“ash is indeed representative of all material being erupted”**. However, the reviewer’s point is valid, as there could be a higher crystallinity effect on our rheological measurements than what we have interpreted; this may indeed explain the discrepancy between

the measured viscosities in the range of 1200 to 1150°C and the predicted GRD (crystal-free values). We have therefore addressed this point by quantifying the crystallinities of six select tephra fragments (please see also our response to reviewer 1's similar comments) and used these textural measurements to apply a rheological model (Klein et al., 2018). Full details of this analysis are given in the response to reviewer 1's comments on the same issue, as well as in the revised manuscript. To summarize, these results indicate that for the analyzed materials, the natural tephra crystallinities (~6-16 vol.%) would translate to *upward* shifts in the viscosity of at most a factor of about two (but mostly between 1.3 to 1.9). These shifts are roughly equal to the observed offsets between our measured and predicted viscosity values shown in Fig. 6. All of this is clarified in the revised text on lines 278-302, which we adjusted in response to Reviewer 1's similar comment. Finally, we have corrected our assessment of 30 vol.% crystallinity in the tephra pictured in Fig. 5 as this was erroneous; the accurate groundmass microlite crystallinity covers approximately the same range as the microphenocryst population, about 5-10 vol.%. This has been corrected in the revised manuscript.

To summarize so far: The physical state of the lava upon eruption remains unknown, but as the point of our paper is to establish permissible rheological conditions of this *newly emergent* basanite through direct viscosity determinations, we have nonetheless established what the lava's rheology could have been, now accounting for effect of crystals. In this vein, and in order to address the reviewer's main request, we can now say yes, **"the increase in measured viscosity relative to the crystal-free value predicted by GRD is comparable"**.

The reviewer also notes the difference between the rheometry heating experiment textures and the natural tephra samples. We agree that this is a marked difference, and have added a bit of clarifying text that proposes that these differences—in particular the relatively lower crystallinities observed in the 1-atm heating experiments—could arise from kinetic limitations to crystal growth, as the time over which the experiments were run was very short (15-30 minutes) and perhaps crystals in the rheology experiments could have continued growing due to the long (hours) nature of those experiments. This said, however, we do not believe that the natural tephra textures (being more crystalline than the short heating experiments) provide a more accurate proxy for what the rheology experiments comprised in terms of mineralogy and texture. This remains equivocal and we consequently refrain from making this point in the revised manuscript. The reasons for the mineralogical differences are furthermore uncertain, but may include the fact that the rheology experiments were conducted at 1-atm, highly oxidizing conditions, and on melts that were effectively anhydrous (0.005-0.02 wt.% H₂O according to new FTIR data provided in the revised Supplementary Information), factors all of which will dictate mineralogy and total crystallinities.

Therefore, to summarize this response, we interpreted the observed crystallinities in heating experiments to represent minimum crystallinities that would likely increase during the course of the long (hours to days) rheology experiments. We

have adjusted the text to account for this interpretation. The modified text appears on lines 251-256 of the revised manuscript:

“These 1-atm experiments furthermore indicate relatively subtle (<0.1 to ~4 vol.%) crystallinities in melts at higher T’s (~1163-1265°C). However, we consider these crystal contents to reflect minimum values given that the short durations of these experiments (10’s of mins) may not have fostered attainment of textural equilibrium at a given T.”

3) For the Re calculations in the discussion, it appears that the authors use only the pure melt viscosity when considering the hydrous melt case? The crystals present in the BSE images in Figure 5 should still be accounted for. If their effect turns out to be negligible overall in the Re calculations, it should still be mentioned.

We appreciate this comment and how the reviewer correctly points out that when dealing with lava *Re* calculations, we have to take into account the effective viscosity as determined by the presence of crystals. As discussed in the detailed response to the reviewer’s 2nd point above, we reiterate that our new relative viscosity calculations that stem from textural measurements on tephra indicate that the viscosity increases due to crystals will be subtle, only a factor of about 1.3 to 1.9. These results will therefore not shift *Re* calculations significantly, but they will result in lower *Re*. In the original manuscript we made the point that *Re* calculations are considered to be accurate to an order of magnitude; this was to account for some but not all possible variation in the many *Re* variables, as well as uncertainties in estimating flow length scales. We have nonetheless modified the *Re* calculations in the revised manuscript to include *effective viscosities* in the *Re* calculations. These indeed show that the influence of crystals on the final *Re* numbers is minor and that turbulent flow is still a valid outcome. The modified paragraph (lines 381-399) reads as follows:

*“The viscosity, μ , is the apparent (or effective) viscosity, in Pa s. As we are utilizing dynamic information gained from remote observations, the flow dimensions (channel width and depth) carry some error; we estimate uncertainties of ± 2 m in channel width and ± 0.2 m in the depth values (Supp. Info.). Due to the irregularity of the channels themselves (Fig. 2), flow depth and width are not constant along the lengths of the cascades. We therefore consider *Re* estimates to be accurate to an order of magnitude. Using an approximate, and minimum 0.3 meter flow depth, a width of 3 m, magma density of 2700 kg m^{-3} (ref. 47), a velocity of 10 m s^{-1} (determined from video analysis of the lava cascades; Fig. 2) and μ of 20 Pa s —an estimate based on a hydrous (~0.8 wt.% H_2O) basanite at an intermediate eruption *T* of 1175°C (Fig. 6) and adjusted for the effect of crystals by way of a feasible relative viscosity of 1.6—yields *Re* of ~1350. Clearly, *Re* will be higher for a lower melt viscosity, e.g., corresponding to a higher eruption temperature (~ 1200°C ; Table 1). Additionally, *Re* will rise for greater characteristic lengths (e.g., flow depth) and velocities. A potential upper limit in *Re* of ~1930 is reached by applying an effective viscosity of 14 Pa s , which is the minimum*

permissible hydrous (0.8 wt.% H₂O) basanite viscosity (Fig. 6; Table 1), and using all other previously defined variables. "

Overall, these points are meant to clarify whether the lava is truly “ultralow viscosity” in terms of bulk viscosity, though certainly the pure melt is quite low in viscosity.

Thanks for this well-made point. We think that the case for ultralow melt and bulk viscosity are now robustly made with the additional calculations of effective viscosity suggested by both this and the other reviewer.

It does not seem like ultralow viscosity is a prerequisite for the observed standing waves as they seem to be quite common in Kīlauea eruptions when effusion rate is high (Le Moigne et al., 2020).

We appreciate this comment. We do not make this point that low viscosity is a prerequisite in the paper. We also do not have the space in our paper to make a thorough comparison of Le Moigne et al.’s observations with flow properties at Cumbre Vieja.

I do find it hard to evaluate the necessity of ultralow viscosity for the other observed flow abnormality (i.e., the propagating wave fronts in the cascades) as I am unaware of any studies noting their presence in other eruptions.

We appreciate this comment. It underscores how important it is to document these hitherto unseen features in the lava cascades. Again, we do not make the reviewer’s suggested point that ultralow viscosities are *prerequisites* or *necessitate* the various flow phenomena. The phenomena, are however, *consistent* with the low viscosities we report. We made this point on line 357 of the original manuscript. In another instance (lines 391-394), we contended that the flow phenomena (standing waves and cascade flows’ behavior) “validate” the experimental evidence for ultralow viscosity. This stems from the fact that low viscosities favor higher Reynolds numbers, which in turn scale directly with the Froude number (ie., $Fr=(Re/G)^{1/2}$ where G is the ratio of buoyancy to viscous stress; ref. 20, Griffiths, 2000). We have added a reference to Griffiths (2000) to the revised sentence (lines 456-460) to support our contention that the flow behavior is entirely consistent with low melt viscosities, in turn indicative of high Re conditions for the relevant flow conditions. The revised sentence reads:

*“In summary, the observations of near-vent flow phenomena on 18 November 2021 in addition to the occurrence of standing waves that formed days later (Fig. 3) validate the experimental evidence for ultralow viscosity of the basanite, **the high calculated Re’s**, and its consequent emplacement as rapidly moving supercritical lava flows²⁰.”*

Even the rapid flow emplacement times are not a “smoking gun” for an ultralow bulk viscosity lava because the emplacement times the authors list for this eruption are comparable to the emplacement times for flows from the

later stages of the Kīlauea 2018 eruption. The flows from fissures 20/22 traveled 5.5 km in ~28 hours (deGraffenried et al., 2021, Bulletin of Volcanology) and the major flow from Ahu'ailā'au (formerly fissure 8) traveled 13 km in 6 days (Dietterich et al., 2021, Bulletin of Volcanology). Many of these factors tie into effusion rate, as well as bulk viscosity, so it is important to ensure which parameter is the cause.

This comment, like the reviewer's other previous two remarks, implies that we have somehow drawn a conclusion that low viscosity is causing various behavior at the exclusion of other parameters. We never claimed this (nor do we consider low viscosity a "smoking gun"), and our text is unambiguous in this regard. We do not assume that viscosity is the only player, but rather that our experimental *measurements* and subsequent (new) calculations thereof (effective μ 's) show unequivocally that this eruption produced mafic magma with a distinctly lower viscosity than typical basaltic eruptions. It is interesting that the reviewer makes examples of two lavas that not only were fed by high fluxes but also were among the lowest viscosity flows of the 2018 Kilauea eruption (e.g., ref. 42, Gransecki et al., 2018). We hesitate to comment further here as it is ultimately difficult to rank the importance of several dependent variables and impossible to "ensure which parameter is the cause". That latter task is furthermore well beyond the scope of this paper.

Line by line comments:

Lines 17-19: if my second general point holds and the crystals do indeed impact viscosity in a non-negligible way, sentences like this one will need to be revised throughout the text as crystallinity does play a role, though not to increase viscosity by increasing melt SiO₂^[L]_[SEP]

Thank you for this comment. We can see how this was confusing. Our intended point was one of the *chemical* shift caused by crystallization (differentiation) and not the physical effects. We have adjusted the sentence in the abstract and the new phrase reads (see lines 17-19):

"Increases in viscosity due to crystallization-induced melt differentiation were subdued in this eruption, due in part to subtle degrees of silica enrichment in alkaline magma."

Line 75: "a elongate" should be "an elongate"

Thank you for this correction. We have made the change.

Line 140: "strea" should be "stream"^[L]_[SEP]

Thank you for this correction. We have made the change.

Line 142: "Kilauea" should be "Kīlauea"

Thank you for this correction. We have made the change.

Line 173: should be “These results”?

Thank you for this correction. We have made the change.

Line 195: missing the opening parentheses before Fig. 4 call

Thank you for this correction. We have made the change.

Line 263: Saying that crystals increase “bulk stress” is somewhat confusing to me, as stress is a force applied to the system. Crystals do change the response to stress, however. This sentence should be revised to be clearer what the authors mean.

We have to disagree with the reviewer here. The way we express the effect of crystals on suspension rheology is entirely correct, and is consistent with and derives from the terminology and analysis of Batchelor (see original ref. 33), who in 1970 so eloquently demonstrated that (spherical) solids, by virtue of having *surfaces* that create a no-slip boundary condition with the suspending fluid, necessarily impart added *stress* on the system and therefore raise viscosity. We have added a more recent reference (ref. 39) that also follows this reasoning and terminology and details how the stress is calculated through integration over all particle surfaces (see new ref. 39, Brady et al., 2006). Stress, furthermore, is defined as force applied to an area (in this case the crystal surfaces). We have left this passage unchanged but thank the reviewer for their inquiry.

Line 278: “conditions that are supported by”

Thank you for this correction. We have made the change.

Lines 280-281: Are these water values representative of water contents at depth or what is still dissolved in the glass upon eruption? Only the latter will impact lava melt viscosity, and 1 wt.% seems awfully high for what is dissolved in a melt upon eruption for a low viscosity melt that should have no trouble degassing.

Thank you for these comments and questions. We envision these values as being realistic for shallow ascending and “freshly emergent” magma; importantly, these values, are corroborated by other studies cited in the manuscript, and define a range of permissible shallow magma H₂O contents in the event that the plagioclase microlites grew during magma ascent and eruption. We have now also added a reference to the loss on ignition (LOI) values in the bulk chemistry data displayed in Table 1 (see line 324 of the revised manuscript), as this also reflects preserved volatile contents in the tephra, particularly H₂O (0.71±0.09 wt.%), that mesh with our other estimates pointing towards upwards of 0.8 wt.% H₂O. It is important to note that we do not have firm constraints on when and where the plagioclase

microlites grew, nor do we have samples of lava to confirm plagioclase was present in lava at the time of its eruption. Nonetheless, we think that our citing of multiple lines of evidence to suggest that the magma was hydrous is a sufficient ground to establish a permissible range of viscosities of magma, which is a main goal of this work. We have left this passage otherwise unchanged in the revised manuscript.

Line 334: How was lava flow depth assessed?^[L]_[SEP]

We appreciate this inquiry. We have clarified (in the revised Supplementary Information) that this estimate derives from analysis of lava flow thickness in a flow that was erupted on 25 November 2021 and documented in online videos of which we do not own the copyright. Because the copyright owner for the original video of this flow is unknown, we cannot include this information in the submission. However, in the revised Supplementary Information, we do show a screenshot of the flow on which we estimated thickness and this indicates what appears to be lava on order of 30 cm thick. We emphasize that this is a rough estimate based on comparisons of the flow with particles in the substrate over which it flowed.

General grammar comment: the authors switch between using an Oxford comma and not – either one is fine, but the authors should be consistent in the style they choose.^[L]_[SEP]

Thank you for this comment. We've gone through the revised draft and corrected where appropriate, the use of commas.

Table 1: it would be useful to list the starting crystal contents of the tephra as well

Thank you for this useful suggestion. We have updated the Table 1 to reflect the approximate crystallinities of the tephra samples.

If any of my points are unclear, the authors are welcome to contact me for clarification!

We are grateful for your thoughtful comments. Thank you.

Sincerely,^[L]_[SEP] **Rebecca deGraffenried**

REVIEWERS' COMMENTS

Reviewer #2 (Remarks to the Author):

I appreciate that the authors have carefully and thoroughly addressed all of my points, as well as answered my questions and clarified the things I misunderstood. I have no further comments on the revised manuscript, and I believe that it is suitable for publication in its current form. Thanks to the authors for submitting this interesting work!